# Experimental Design for Overparameterized Learning with Application to Single Shot Deep Active Learning

## Abstract

The impressive performance exhibited by modern machine learning models hinges on the ability to train such models on a very large amounts of labeled data. However, since access to large volumes of labeled data is often limited or expensive, it is desirable to alleviate this bottleneck by carefully curating the training set. Optimal experimental design is a well-established paradigm for selecting data point to be labeled so to maximally inform the learning process. Unfortunately, classical theory on optimal experimental design focuses on selecting examples in order to learn underparameterized (and thus, non-interpolative) models, while modern machine learning models such as deep neural networks are overparameterized, and oftentimes are trained to be interpolative. As such, classical experimental design methods are not applicable in many modern learning setups. Indeed, the predictive performance of underparameterized models tends to be variance dominated, so classical experimental design focuses on variance reduction, while the predictive performance of overparameterized models can also be, as is shown in this paper, bias dominated or of mixed nature. In this paper we propose a design strategy that is well suited for overparameterized regression and interpolation, and we demonstrate the applicability of our method in the context of deep learning by proposing a new algorithm for single shot deep active learning.

## 1 Introduction

The impressive performance exhibited by modern machine learning models hinges on the ability to train the aforementioned models on a very large amounts of labeled data. In practice, in many real world scenarios, even when raw data exists aplenty, acquiring labels might prove challenging and/or expensive. This severely limits the ability to deploy machine learning capabilities in real world applications. This bottleneck has been recognized early on, and methods to alleviate it have been suggested. Most relevant for our work is the large body of research on *active learning* or *optimal experimental design*, which aims at selecting data point to be labeled so to maximally inform the learning process. Disappointedly, active learning techniques seem to deliver mostly lukewarm benefits in the context of deep learning.

One possible reason why experimental design has so far failed to make an impact in the context of deep learning is that such models are *overparameterized*, and oftentimes are trained to be *interpolative* (Zhang et al., 2017), i.e., they are trained so that a perfect fit of the training data is found. This raises a conundrum: the classical perspective on statistical learning theory is that overfitting should be avoided since there is a tradeoff between the fit and complexity of the model. This conundrum is exemplified by the *double descent phenomena* (Belkin et al., 2019b; Bartlett et al., 2020), namely when fixing the model size and increasing the amount of training data, the predictive performance initially goes down, and then starts to go up, exploding when the amount of training data approaches the model complexity, and then starts to descend again. This runs counter to statistical intuition which says that more data implies better learning. Indeed, when using interpolative models, more data can hurt (Nakkiran et al., 2020a)! This phenomena is exemplified in the curve labeled "Random Selection" in Figure 1. Figure 1 explores the predictive performance of various designs when learning a linear regression model and varying the amount of training data with responses.

The fact that more data can hurt further motivates experimental design in the interpolative regime. Presumably, if data is carefully curated, more data should never hurt. Unfortunately, classical optimal experimental design focuses on the underparameterized (and thus, non-interpolative) case. As such, the theory reported in the literature is often not applicable in the interpolative regime. As our analysis shows (see Section 3), the prediction error of interpolative models can either be *bias dominated* (the first descent phase, i.e., when training size is very small compared to the number of parameters), *variance dominated* (near equality of size and parameters) or of *mixed nature*. However, properly trained underparameterized models tend to have prediction error which is variance dominated, so classical experimental design focuses on variance reduction. As such, naively using classical optimality criteria, such as V-optimality (the one most relevant for generalization error) or others, in the context of interpolation, tends to produce poor results when prediction error is bias dominated or of mixed nature. This is exemplified in the curve labeled "Classical OED" in Figure 1.

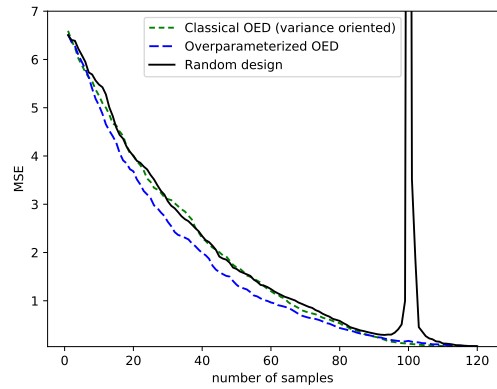

Figure 1: MSE of a minimum norm linear interpolative model. We use synthetic data of dimension 100. The full description is in Appendix E.

The goal of this paper is to understand these regimes, and to propose an experimental design strategy that is well suited for overparameterized models. Like many recent work that attempt to understand the double descent phenomena by analyzing underdetermined linear regression, we too use a simple linear regression model in our analysis of experimental design in the overparameterized case (however, we also consider kernel ridge regression, not only linear interpolative models). We believe that understanding experimental design in the overparameterized linear regression case is a prelude to designing effective design algorithms for deep learning. Indeed, recent theoretical results showed a deep connection between deep learning and kernel learning via the so-called Neural Tangent Kernel (Jacot et al., 2018; Arora et al., 2019a; Lee et al., 2019). Based on this connection, and as a proof-of-concept, we propose a new algorithm for single shot deep active learning.

Let us now summarize our contributions:

- We analyze the prediction error of learning overparameterized linear models for a given fixed design, revealing three possible regimes that call for different design criteria: bias dominated, variance dominated, and mixed nature. We also reveal an interesting connection between overparameterized experimental design and the *column subset selection problem* (Boutsidis et al., 2009), *transductive experimental design* (Yu et al., 2006), and *coresets* (Sener & Savarese, 2018). We also extend our approach to kernel ridge regression.
- We propose a novel greedy algorithm for finding designs for overparameterized linear models. As exemplified in the curve labeled "Overparameterized OED", our algorithm is sometimes able to mitigate the double descent phenomena, while still performing better than classical OED (though no formal proof of this fact is provided).
- We show how our algorithm can also be applied for kernel ridge regression, and report experiments which show that when the number of parameters is in a sense infinite, our algorithm is able to find designs that are better than state of the art.
- We propose a new algorithm for single shot deep active learning, a scarcely treated problem so far, and demonstrate its effectiveness on MNIST.

**Related Work.** The phenomena of benign overfitting and double descent was firstly recognized in DNNs (Zhang et al., 2017), and later discussed and analyzed in the context of linear models (Zhang et al., 2017; Belkin et al., 2018; 2019a;b; Bartlett et al., 2020). Recently there is also a growing interest in the related phenomena of "more data can hurt" (Nakkiran et al., 2020a; Nakkiran, 2019; Nakkiran et al., 2020b; Loog et al., 2019). A complementary work discussed the need to consider zero or negative regularization coefficient for large real life linear models (Kobak et al., 2020).

Experimental design is an well established paradigm in statistics, extensively covered in the literature for the linear case (Pukelsheim, 2006) and the non linear case (Pronzato & Pázman, 2013). The application of it to pool based active learning with batch acquisitions was explored by Yu et al. (2006) for linear models and by Hoi et al. (2006) for logistic regression. It was also proposed in the context of deep learning (Sourati et al., 2018). Another related line of work is recent work by Haber and Horesh on experimental design for ill-posed inverse problems(Haber et al., 2008; 2012; Horesh et al., 2010). Active learning in the context of overparameterized learning was explored by Karzand & Nowak (2020), however their approach differs from ours significantly since it is based on artificially completing the labels using a minimax approach.

I the context of *Laplacian regularized Least Squares* (LapRLS), which is a generalization of ridge regression, Gu et al. (2012) showed rigorously that Yu et al. (2006) criterion is justified as a bound for both the bias and variance components of the expected error. We farther show that this bound is in some sense tight only if the parameter norm is oneand the noise variance equals the $l_2$ penalty coefficient. In addition we postulate and show experimentally that in the overparameterized case using a bias dominant criterion is preferable. Another case in which the bias term idoes not vanish is when the model is misspecified. For linear and generalized linear models this case has been tackled with reweighing of the loss function.

A popular modern approach for pool based active learning with batch acquisition is coresets (Sener & Savarese, 2018; Geifman & El-Yaniv, 2017; Ash et al., 2019; Pinsler et al., 2019). This approach has been used in the context of active learning for DNNs.

## 2 UNDERPARAMETERIZED V-OPTIMAL EXPERIMENTAL DESIGN

Consider a noisy linear response model $y = \mathbf{x}^T\mathbf{w} + \epsilon$, where $\epsilon \sim \mathcal{N}(0, \sigma^2)$ and $\mathbf{w} \in \mathbb{R}^d$ and assume we are given with some data points $\mathbf{x}_1, \ldots, \mathbf{x}_n$, for which we obtained independent responses, $y_i = \mathbf{x}_i^T\mathbf{w} + \epsilon_i$. Consider the underparameterized case, i.e. $n \geq d$, and furthermore assume that the set $\{\mathbf{x}_1, \ldots, \mathbf{x}_n\}$ contains at least $d$ independent vectors. The best linear unbiased estimator $\hat{\mathbf{w}}$ of $\mathbf{w}$ according to the Gauss-Markov theorem is given by: $\hat{\mathbf{w}} = \arg\min_{\mathbf{w}} \|\mathbf{X}\mathbf{w} - \mathbf{y}\|_2^2 = \mathbf{X}^+\mathbf{y}$ where $\mathbf{X} \in \mathbb{R}^{n \times d}$ is a matrix whose rows are $\mathbf{x}_1, \ldots, \mathbf{x}_n$, $\mathbf{y} = [y_1 \ldots y_n]^T \in \mathbb{R}^n$ and $\mathbf{X}^+$ is the Moore-Pensrose pseudoinverse of $\mathbf{X}$. It is well known that $\hat{\mathbf{w}} - \mathbf{w}$ is a normal random vector with zero mean and covariance matrix $\sigma^2\mathbf{M}^{-1}$, where $\mathbf{M} = \mathbf{X}^T\mathbf{X}$ is the Fisher information matrix. This implies that $\hat{y}(\mathbf{x}) - y(\mathbf{x})$ is also a normal variable with zero mean and variance equal to $\sigma^2\mathbf{x}^T\mathbf{M}^{-1}\mathbf{x}$.

Assume also that $\mathbf{x}$ comes from a distribution $\rho$. With that we can further define the excess risk $R(\hat{\mathbf{w}}) = \mathbb{E}_{\mathbf{x}\sim\rho}\left[(\mathbf{x}^T\mathbf{w} - \mathbf{x}^T\hat{\mathbf{w}})^2\right]$ and its expectation:

$$\mathbb{E}_\epsilon\left[R(\hat{\mathbf{w}})\right] = \mathbb{E}_{\mathbf{x}\sim\rho}\left[\text{Var}_\epsilon\left[y(\mathbf{x}) - \hat{y}(\mathbf{x})\right]\right] = \mathbb{E}_{\mathbf{x}\sim\rho}\left[\sigma^2\mathbf{x}^T\mathbf{M}^{-1}\mathbf{x}\right] = \mathbf{Tr}\left(\sigma^2\mathbf{M}^{-1}\mathbf{C}_\rho\right) \quad (1)$$

where $\mathbf{C}_\rho$ is the uncentered second moment matrix of $\rho$: $\mathbf{C}_\rho \coloneqq \mathbb{E}_{\mathbf{x}\sim\rho}\left[\mathbf{x}\mathbf{x}^T\right]$.

Eq. (1) motivates the so-called *V-optimal design* criterion: select the dataset $\mathbf{x}_1, \ldots, \mathbf{x}_n$ so that $\varphi(\mathbf{M}) \coloneqq \mathbf{Tr}\left(\mathbf{M}^{-1}\mathbf{C}_\rho\right)$ is minimized (if we do not have access to $\mathbf{C}_\rho$ then it is possible to estimate it by drawing samples from $\rho$). In doing so, we are trying to minimize the expected (with respect to the noise $\epsilon$) average (with respect to the data $\mathbf{x}$) prediction variance, since the risk is composed solely from it (due to the fact that the estimator is unbiased). As we shall see, this is in contrast with the overparameterized case, in which the estimator is biased.

V-optimality is only one instance of various statistical criteria used in experimental design. In general experimental design, the focus is on minimizing a preselected criteria $\varphi(\mathbf{M})$ (Pukelsheim, 2006). For example in D-optimal design, $\varphi(\mathbf{M}) = \det(\mathbf{M}^{-1})$ and in A-optimal design $\varphi(\mathbf{M}) = \mathbf{Tr}\left(\mathbf{M}^{-1}\right)$. However, since minimizing the V-optimality criterion corresponds to minimizing the risk, it is more appropriate when assessing the predictive performance of machine learning models.

## 3 OVERPARAMETERIZED EXPERIMENTAL DESIGN CRITERIA

In this section we derive an expression for the risk in the overparameterized case, i.e. like Eq. (1) but also for the case that $n \leq d$ (our expressions also hold for $n > d$). This, in turn, leads to an

experimental design criteria analogous to V-optimality, but relevant for overparamterized modeling as well. We design a novel algorithm based on this criteria in subsequent sections.

### 3.1 OVERPARAMETERIZED REGRESSION AND INTERPOLATION

When $n \geq d$ there is a natural candidate for $\hat{\mathbf{w}}$: the best unbiased linear estimator $\mathbf{X}^{+}\mathbf{y}$[1]. However, when $d > n$ there is no longer a unique minimizer of $\|\mathbf{X}\mathbf{w} - \mathbf{y}\|_2^2$ as there is an infinite amount of interpolating $\mathbf{w}$'s, i.e. $\mathbf{w}$'s such that $\mathbf{X}\mathbf{w} = \mathbf{y}$ (the last statement makes the mild additional assumption that $\mathbf{X}$ has full row rank). One natural strategy for dealing with the non-uniqueness is to consider the *minimum norm interpolator*:

$$\hat{\mathbf{w}} := \arg\min \|\mathbf{w}\|_2^2 \text{ s.t. } \mathbf{X}\mathbf{w} = \mathbf{y}$$

It is still the case that $\hat{\mathbf{w}} = \mathbf{X}^{+}\mathbf{y}$. Another option for dealing with non-uniqueness of the minimizer is to add a ridge term, i.e., add and additive penalty $\lambda\|\mathbf{w}\|_2^2$. Let:

$$\hat{\mathbf{w}}_\lambda := \arg\min \|\mathbf{X}\mathbf{w} - \mathbf{y}\|_2^2 + \lambda\|\mathbf{w}\|_2^2$$

One can show that

$$\hat{\mathbf{w}}_\lambda = \mathbf{X}_\lambda^{+}\mathbf{y} \tag{2}$$

where for $\lambda \geq 0$ we define $\mathbf{X}_\lambda^{+} := \left(\mathbf{X}^{\mathsf{T}}\mathbf{X} + \lambda\mathbf{I}_d\right)^{+}\mathbf{X}^{\mathsf{T}}$ (see also Bardow (2008)). Note that Eq. (2) holds both for the overparameterized ($d \geq n$) and underparameterized ($d < n$) case.

**Proposition 1.** *The function* $\lambda \mapsto \mathbf{X}_\lambda^{+}$ *is continuous for all* $\lambda \geq 0$.

The proof, like all of our proofs, is delegated to the appendix. Thus, we also have that the minimum norm interpolator $\hat{\mathbf{w}}$ is equal to $\hat{\mathbf{w}}_0$, and that $\lambda \mapsto \hat{\mathbf{w}}_\lambda$ is continuous. This implies that the various expressions for the expected risk of $\hat{\mathbf{w}}_\lambda$ hold also when $\lambda = 0$. So, henceforth we analyze the expected risk of $\hat{\mathbf{w}}_\lambda$ and the results also apply for $\hat{\mathbf{w}}$.

### 3.2 EXPECTED RISK OF $\hat{\mathbf{w}}_\lambda$

The following proposition gives an expression for the expected risk of the regularized estimator $\hat{\mathbf{w}}_\lambda$. Note that it holds both for the overparameterized ($d \geq n$) and underparameterized ($d < n$) case.

**Proposition 2.** *We have*

$$\mathbb{E}\left[R(\hat{\mathbf{w}}_\lambda)\right] = \underbrace{\|\mathbf{C}_\rho^{1/2}\left(\mathbf{I} - \mathbf{M}_\lambda^{+}\mathbf{M}\right)\mathbf{w}\|_2^2}_{bias} + \underbrace{\sigma^2\mathbf{Tr}\left(\mathbf{C}_\rho\mathbf{M}_\lambda^{+2}\mathbf{M}\right)}_{variance}$$

*where* $\mathbf{M}_\lambda := \mathbf{X}^{\mathsf{T}}\mathbf{X} + \lambda\mathbf{I}_d = \mathbf{M} + \lambda\mathbf{I}_d$. *The expectation is with respect to the training noise* $\epsilon$.

The last proposition motivates the following design criterion, which can be viewed as a generalization of classical V-optimality:

$$\varphi_\lambda(\mathbf{M}) := \|\mathbf{C}_\rho^{1/2}\left(\mathbf{I} - \mathbf{M}_\lambda^{+}\mathbf{M}\right)\mathbf{w}\|_2^2 + \sigma^2\mathbf{Tr}\left(\mathbf{C}_\rho\mathbf{M}_\lambda^{+2}\mathbf{M}\right).$$

For $\lambda = 0$ the expression simplifies to the following expression:

$$\varphi_0\left(\mathbf{M}\right) = \|\mathbf{C}_\rho^{1/2}\left(\mathbf{I} - \mathbf{P}_\mathbf{M}\right)\mathbf{w}\|_2^2 + \sigma^2\mathbf{Tr}\left(\mathbf{C}_\rho\mathbf{M}^{+}\right)$$

where $\mathbf{P}_\mathbf{M} = \mathbf{M}^{+}\mathbf{M}$ is the projection on the row space of $\mathbf{X}$. Note that when $n \geq d$ and $\mathbf{X}$ has full column rank, $\varphi_0(\mathbf{M})$ reduces to the variance of underparameterized linear regression, so minimizing $\varphi_\lambda(\mathbf{M})$ is indeed a generalization of the V-optimality criterion.

Note the bias-variance tradeoff in $\varphi_\lambda(\mathbf{M})$. When the bias term is much larger than the variance, something we should expect for small $n$, then it make sense for the design algorithm to be bias oriented. When the variance is larger, something we should expect for $n \approx d$ or $n \geq d$, then the design algorithm should be variance oriented. It is also possible to have mixed nature in which both bias and variance are of the same order.

---

[1] In practice, when $n$ is only mildly bigger than $d$ it is usually better to regularize the problem.

### 3.3 Practical Criterion

As is, $\varphi_\lambda$ is problematic as an experimental design criterion since it depends both on $\mathbf{w}$ and on $\mathbf{C}_\rho$. We discuss how to handle an unknown $\mathbf{C}_\rho$ in Subsection 3.5. Here we discuss how to handle an unknown $\mathbf{w}$. Note that obviously $\mathbf{w}$ is unknown: it is exactly what we want to approximate! If we have a good guess $\tilde{\mathbf{w}}$ for the true value of $\mathbf{w}$, then we can replace $\mathbf{w}$ with $\tilde{\mathbf{w}}$ in $\varphi_\lambda$. However, in many cases, such an approximation is not available. Instead, we suggest to replace the bias component with an upper bound: $\|\mathbf{C}_\rho^{1/2}\left(\mathbf{I} - \mathbf{M}_\lambda^+\mathbf{M}\right)\mathbf{w}\|_2^2 \leq \|\mathbf{w}\|_2^2 \cdot \|\mathbf{C}_\rho^{1/2}\left(\mathbf{I} - \mathbf{M}_\lambda^+\mathbf{M}\right)\|_F^2$.

Let us now define a new design criterion which has an additional parameter $t \geq 0$:

$$\bar{\varphi}_{\lambda,t}(\mathbf{M}) = \underbrace{\|\mathbf{C}_\rho^{1/2}\left(\mathbf{I} - \mathbf{M}_\lambda^+\mathbf{M}\right)\|_F^2}_{\text{bias bound (divided by }\|\mathbf{w}\|_2^2)} + \underbrace{t\mathbf{Tr}\left(\mathbf{C}_\rho\mathbf{M}_\lambda^{+^2}\mathbf{M}\right)}_{\text{variance (divided by }\|\mathbf{w}\|_2^2)}.$$

The parameter $t$ captures an a-priori assumption on the tradeoff between bias and variance: if we have $t = \sigma^2/\|\mathbf{w}\|_2^2$, then $\varphi_\lambda(\mathbf{M}) \leq \|\mathbf{w}\|_2^2 \cdot \bar{\varphi}_{\lambda,t}(\mathbf{M})$. Thus, minimizing $\bar{\varphi}_{\lambda,t}(\mathbf{M})$ corresponds to minimizing an upper bound of $\varphi_\lambda$, if $t$ is set correctly.

Another interpretation of $\bar{\varphi}_{\lambda,t}(\mathbf{M})$ is as follows. If we assume that $\mathbf{w} \sim \mathcal{N}(0, \gamma^2\mathbf{I}_d)$, then

$$\mathbb{E}_\mathbf{w}\left[\varphi_\lambda(\mathbf{M})\right] = \gamma^2\|\mathbf{C}_\rho^{1/2}\left(\mathbf{I} - \mathbf{M}_\lambda^+\mathbf{M}\right)\|_F^2 + \sigma^2\mathbf{Tr}\left(\mathbf{C}_\rho\mathbf{M}_\lambda^{+^2}\mathbf{M}\right)$$

so if we set $t = \sigma^2/\gamma^2$ then $\gamma^2\bar{\varphi}_{\lambda,t}(\mathbf{M}) = \mathbb{E}_\mathbf{w}\left[\varphi_\lambda(\mathbf{M})\right]$, so minimizing $\bar{\varphi}_{\lambda,t}(\mathbf{M})$ corresponds to minimizing the expected expected risk if $t$ is set correctly. Again, the parameter $t$ captures an a-priori assumption on the tradeoff between bias and variance.

*Remark* 1. One alternative strategy for dealing with the fact that $\mathbf{w}$ is unknown is to consider a sequential setup where batches are acquired incrementally based on increasingly refined approximations of $\mathbf{w}$. Such a strategy falls under the heading of Sequential Experimental Design. In this paper, we focus on *single shot* experimental design, i.e. examples are chosen to be labeled once. We leave sequential experimental design to future research. Although, we decided to focus on the single shot scenario for simplicity, the single shot scenario actually captures important real-life scenarios.

### 3.4 Comparison to Other Generalized V-Optimality Criteria

Consider the case of $\lambda = 0$. Note that we can write: $\bar{\varphi}_{0,t}(\mathbf{M}) = \|\mathbf{C}_\rho^{1/2}\left(\mathbf{I} - \mathbf{P_M}\right)\|_F^2 + t\mathbf{Tr}\left(\mathbf{C}_\rho\mathbf{M}^+\right)$. Recall that the classical V-optimal experimental design criterion is $\mathbf{Tr}\left(\mathbf{C}_\rho\mathbf{M}^{-1}\right)$, which is only applicable if $n \geq d$ (otherwise, $\mathbf{M}$ is not invertible). Indeed, if $n \geq d$ and $\mathbf{M}$ is invertible, then $\mathbf{P_M} = \mathbf{I}_d$ and $\bar{\varphi}_{0,t}(\mathbf{M})$ is equal to $\mathbf{Tr}\left(\mathbf{C}_\rho\mathbf{M}^{-1}\right)$ up to a constant factor. However, $\mathbf{M}$ is not invertible if $n < d$ and the expression $\mathbf{Tr}\left(\mathbf{C}_\rho\mathbf{M}^{-1}\right)$ does not make sense.

One naive generalization of classical V-optimality for $n < d$ would be to simply replace the inverse with pseudoinverse, i.e. $\mathbf{Tr}\left(\mathbf{C}_\rho\mathbf{M}^+\right)$. This corresponds to minimizing only the variance term, i.e. taking $t \to \infty$. This is consistent with classical experimental design which focuses on variance reduction, and is appropriate when the risk is variance dominated.

Another generalization of V-optimality can be obtained by replacing $\mathbf{M}$ with its regularized (and invertible) version $\mathbf{M}_\mu = \mathbf{M} + \mu\mathbf{I}_d$ for some chosen $\mu > 0$, obtaining $\mathbf{Tr}\left(\mathbf{C}_\rho\mathbf{M}_\mu^{-1}\right)$. This is exactly the strategy employed in transductive experimental design (Yu et al., 2006), and it also emerges in a Bayesian setup (Chaloner & Verdinelli, 1995). One can try to eliminate the parameter $\mu$ by taking the limit of the minimizers when $\mu \to 0$. The following proposition shows that this is actually equivalent to taking $t = 0$.

**Proposition 3.** *For a compact domain $\Omega \subset \mathbb{R}^{d \times d}$ of symmetric positive semidefinite matrices:*

$$\lim_{\mu \to 0} \underset{\mathbf{M} \in \Omega}{\operatorname{argmin}} \mathbf{Tr}\left(\mathbf{C}_\rho\mathbf{M}_\mu^{-1}\right) \subseteq \underset{\mathbf{M} \in \Omega}{\operatorname{argmin}} \mathbf{Tr}\left(\mathbf{C}_\rho\left(\mathbf{I} - \mathbf{P_M}\right)\right).$$

We see that the aforementioned generalizations of V-optimality correspond to either disregarding the bias term ($t = \infty$) or disregarding the variance term ($t = 0$). However, using $\bar{\varphi}_{0,t}(\mathbf{M})$ allows much better control over the bias-variance tradeoff (see Figure 1.)

Let us consider now the case of $\lambda > 0$. We now show that the regularized criteria $\mathbf{Tr}\left(\mathbf{C}_\rho\mathbf{M}_\mu^{-1}\right)$ used in transductive experimental design (See Proposition 3) when $\mu = \lambda$ corresponds to also using $t = \lambda$.

**Proposition 4.** *For any matrix space* $\Omega$, $\lambda > 0$: $\operatorname{argmin}_{\mathbf{X} \in \Omega} \mathbf{Tr}\left(\mathbf{C}_\rho \mathbf{M}_\lambda^{-1}\right) = \operatorname{argmin}_{\mathbf{X} \in \Omega} \bar{\varphi}_{\lambda,\lambda}(\mathbf{M})$

So, transductive experimental design corresponds to a specific choice of bias-variance tradeoff. Another interesting relation with transductive experimental design is given by next proposition which is a small modification of Theorem 1 due to Gu et al. (2012) .

**Proposition 5.** *For any* $\lambda > 0$ *and* $t \geq 0$: $\bar{\varphi}_{\lambda,t}(\mathbf{M}) \leq (\lambda + t) \mathbf{Tr}\left(\mathbf{C}_\rho \mathbf{M}_\lambda^{-1}\right)$

In the absence of a decent model of the noise, which is a typical situation in machine learning, Prop. 5 suggests the to perhaps minimize only $\mathbf{Tr}\left(\mathbf{C}_\rho \mathbf{M}_\lambda^{-1}\right)$ without need to set $t$. However, this approach may be suboptimal in the overparameterized regime. This approach implicitly considers $t = \lambda$ (see Prop. 4) which in a bias dominated regime can put too much emphasis on minimizing the variance. A sequential approach for experimental design can lead to better modeling of the noise, thereby assisting in dynamically setting $t$ during acquisition-learning cycles. However, in a single shot regime, noise estimation is difficult. Arguably, there exists better values for $t$ than using a default rule-of-thumb $t = \lambda$. In particular, we conjecture that $t = 0$ is a better rule-of-thumb then $t = \lambda$ for severely overparameterized regimes as it suppresses the potential damage of choosing a too large $\lambda$ and it is reasonale also if $\lambda$ is small (since anyway we are in a bias dominated regime), so we can focus on minimizing the bias only. In the experiment section we show an experiment that supports this assumption. Notice that $t = \infty$ corresponds to minimizing the variance, while $t = 0$ corresponds to minimizing the bias.

## 3.5 APPROXIMATING $\mathbf{C}_\rho$

Our criteria so far depended on $\mathbf{C}_\rho$. Oftentimes $\mathbf{C}_\rho$ is unknown. However, it can be approximated using unlabeled data. Suppose we have $m$ unlabeled points (i.e. drawn form $\rho$), and suppose we write them as the rows of $\mathbf{V} \in \mathbb{R}^{m \times d}$. Then $\mathbb{E}\left[m^{-1}\mathbf{V}^\mathsf{T}\mathbf{V}\right] = \mathbf{C}_\rho$. Thus, we can write

$$m\varphi_\lambda(\mathbf{M}) \approx \psi_\lambda(\mathbf{M}) := \|\mathbf{V}\left(\mathbf{I}_d - \mathbf{M}_\lambda^+\mathbf{M}\right)\mathbf{w}\|_2^2 + \sigma^2 \mathbf{Tr}\left(\mathbf{V}\mathbf{M}_\lambda^{+^2}\mathbf{M}\mathbf{V}^\mathsf{T}\right), \quad \lambda \geq 0.$$

and use $\psi_\lambda(\mathbf{M})$ instead of $\varphi_\lambda(\mathbf{M})$. For minimum norm interpolation we have

$$\psi_0(\mathbf{M}) = \|\mathbf{V}\left(\mathbf{I}_d - \mathbf{P}_\mathbf{M}\right)\mathbf{w}\|_2^2 + \sigma^2 \mathbf{Tr}\left(\mathbf{V}\mathbf{M}^+\mathbf{V}^\mathsf{T}\right).$$

Again, let us turn this into a practical design criteria by introducing an additional parameter $t$:

$$\bar{\psi}_{\lambda,t}(\mathbf{M}) := \|\mathbf{V}\left(\mathbf{I}_d - \mathbf{M}_\lambda^+\mathbf{M}\right)\|_F^2 + t\,\mathbf{Tr}\left(\mathbf{V}\mathbf{M}_\lambda^{+^2}\mathbf{M}\mathbf{V}^\mathsf{T}\right). \tag{3}$$

## 4 POOL-BASED OVERPARAMETERIZED EXPERIMENTAL DESIGN

In the previous section we defined design criteria $\bar{\varphi}_{\lambda,t}$ and $\bar{\psi}_{\lambda,t}$ that are appropriate for overparameterized linear regression. While one can envision a situation in which such we are free to choose $\mathbf{X}$ so to minimize the design criteria, in much more realistic pool-based active learning we assume that we are given in advance a large pool of unlabeled data $\mathbf{x}_1, \ldots, \mathbf{x}_m$. The training set is chosen to be a subset of the pool. This subset is then labeled, and learning performed. The goal of pool-based experimental design algorithms is to chose the subset to be labeled.

We formalize the pool-based setup as follows. Recall that to approximate $\mathbf{C}_\rho$ we assumed we have a pool of unlabeled data written as the rows of $\mathbf{V} \in \mathbb{R}^{m \times d}$. We assume that $\mathbf{V}$ serves also as the pool of samples from which $\mathbf{X}$ is selected. For a matrix $\mathbf{A}$ and index sets $\mathcal{S} \subseteq [n]$, $\mathcal{T} \subseteq [d]$, let $\mathbf{A}_{\mathcal{S},\mathcal{T}}$ be the matrix obtained by restricting to the rows whose index is in $\mathcal{S}$ and the columns whose index is in $\mathcal{T}$. If : appears instead of an index set, that denotes the full index set corresponding to that dimension. Our goal is to select a subset $\mathcal{S}$ of cardinality $n$ such that $\bar{\psi}_{\lambda,t}(\mathbf{V}_{\mathcal{S},:}^\mathsf{T}\mathbf{V}_{\mathcal{S},:})$ is minimized (i.e., setting $\mathbf{X} = \mathbf{V}_{\mathcal{S},:}$). Formally, we pose following problem:

**Problem 1.** (Pool-based Overparameterized V-Optimal Design) Given a pool of unlabeled examples $\mathbf{V} \in \mathbb{R}^{m \times d}$, a regularization parameter $\lambda \geq 0$, a bias-variance tradeoff parameter $t \geq 0$, and a design size $n$, find a minimizer of

$$\min_{\mathcal{S} \subseteq [m],\, |\mathcal{S}| = n} \bar{\psi}_{\lambda,t}(\mathbf{V}_{\mathcal{S},:}^\mathsf{T}\mathbf{V}_{\mathcal{S},:}).$$

Problem 1 is a generalization of the Column Subset Selection Problem (CSSP) (Boutsidis et al., 2009). In the CSSP, we are given matrix $\mathbf{U} \in \mathbb{R}^{d \times m}$ and target number of columns $n$, and our goal is to select a subset $\mathcal{T}$ which is a minimizer of

$$\min_{\mathcal{T} \subseteq [m], |\mathcal{T}| = n} \|(\mathbf{I}_d - \mathbf{U}_{:,\mathcal{T}} \mathbf{U}_{:,\mathcal{T}}^+)\mathbf{U}\|_F^2$$

When $\lambda = 0$ and $t = 0$, Problem 1 reduces to the CSSP for $\mathbf{U} = \mathbf{V}^{\mathrm{T}}$. The $\lambda = t = 0$ case is also somewhat related to the *coreset* approach for active learning Sener & Savarese (2018); Pinsler et al. (2019); Ash et al. (2019); Geifman & El-Yaniv (2017). See Appendix B.

## 5  OPTIMIZATION ALGORITHM

In this section we propose an algorithm for overparameterized experimental design. Our algorithm is based on greedy minimization of a kernalized version of $\bar{\psi}_{\lambda,t}(\mathbf{V}_{\mathcal{S},:}^{\mathrm{T}} \mathbf{V}_{\mathcal{S},:})$. Thus, before presenting our algorithm, we show how to handle feature spaces defined by a kernel.

**Kernelization.** If $|\mathcal{S}| \leq d$ and $\mathbf{V}_{\mathcal{S},:}$ has full row rank we have $(\mathbf{V}_{\mathcal{S},:})_{\lambda}^+ = \mathbf{V}_{\mathcal{S},:}^{\mathrm{T}} (\mathbf{V}_{\mathcal{S},:} \mathbf{V}_{\mathcal{S},:}^{\mathrm{T}} + \lambda \mathbf{I}_{|\mathcal{S}|})^{-1}$ which allows us to write

$$
\begin{aligned}
\bar{\psi}_{\lambda,t}(\mathbf{V}_{\mathcal{S},:}^{\mathrm{T}} \mathbf{V}_{\mathcal{S},:}) &= \mathbf{Tr}\left(\mathbf{V}\left[\mathbf{I} - 2\mathbf{V}_{\mathcal{S},:}^{\mathrm{T}}\left(\mathbf{V}_{\mathcal{S},:}\mathbf{V}_{\mathcal{S},:}^{\mathrm{T}} + \lambda\mathbf{I}_{|\mathcal{S}|}\right)^{-1}\mathbf{V}_{\mathcal{S},:}\right]\mathbf{V}^{\mathrm{T}}\right) \\
&+ \mathbf{Tr}\left(\mathbf{V}\mathbf{V}_{\mathcal{S},:}^{\mathrm{T}}\left(\mathbf{V}_{\mathcal{S},:}\mathbf{V}_{\mathcal{S},:}^{\mathrm{T}} + \lambda\mathbf{I}_{|\mathcal{S}|}\right)^{-1}\mathbf{V}_{\mathcal{S},:}\mathbf{V}_{\mathcal{S},:}^{\mathrm{T}}\left(\mathbf{V}_{\mathcal{S},:}\mathbf{V}_{\mathcal{S},:}^{\mathrm{T}} + \lambda\mathbf{I}_{|\mathcal{S}|}\right)^{-1}\mathbf{V}_{\mathcal{S},:}\mathbf{V}^{\mathrm{T}}\right) \\
&+ t\mathbf{Tr}\left(\mathbf{V}\mathbf{V}_{\mathcal{S},:}^{\mathrm{T}}\left(\mathbf{V}_{\mathcal{S},:}\mathbf{V}_{\mathcal{S},:}^{\mathrm{T}} + \lambda\mathbf{I}_{|\mathcal{S}|}\right)^{-2}\mathbf{V}_{\mathcal{S},:}\mathbf{V}^{\mathrm{T}}\right)
\end{aligned}
$$

Let now $\mathbf{K} := \mathbf{V}\mathbf{V}^{\mathrm{T}} \in \mathbb{R}^{m \times m}$. Then $\mathbf{V}_{\mathcal{S},:}\mathbf{V}_{\mathcal{S},:}^{\mathrm{T}} = \mathbf{K}_{\mathcal{S},\mathcal{S}}$ and $\mathbf{V}\mathbf{V}_{\mathcal{S},:}^{\mathrm{T}} = \mathbf{K}_{:,\mathcal{S}}$. Since $\mathbf{Tr}(\mathbf{K})$ is constant, minimizing $\bar{\psi}_{\lambda,t}(\mathbf{V}_{\mathcal{S},:}^{\mathrm{T}} \mathbf{V}_{\mathcal{S},:})$ is equivalent to minimizing

$$J_{\lambda,t}(\mathcal{S}) := \mathbf{Tr}\left(\mathbf{K}_{:,\mathcal{S}}\left[\left(\mathbf{K}_{\mathcal{S},\mathcal{S}} + \lambda\mathbf{I}_{|\mathcal{S}|}\right)^{-1}\left(-2\mathbf{I}_{|\mathcal{S}|} + \mathbf{K}_{\mathcal{S},\mathcal{S}}\left(\mathbf{K}_{\mathcal{S},\mathcal{S}} + \lambda\mathbf{I}_{|\mathcal{S}|}\right)^{-1}\right) + t\left(\mathbf{K}_{\mathcal{S},\mathcal{S}} + \lambda\mathbf{I}_{|\mathcal{S}|}\right)^{-2}\right]\mathbf{K}_{:,\mathcal{S}}^{\mathrm{T}}\right). \tag{4}$$

For $\lambda = 0$ we have a simpler form: $J_{0,t}(\mathcal{S}) = \mathbf{Tr}\left(\mathbf{K}_{:,\mathcal{S}}\left[-\mathbf{K}_{\mathcal{S},\mathcal{S}}^{-1} + t\mathbf{K}_{\mathcal{S},\mathcal{S}}^{-2}\right]\mathbf{K}_{:,\mathcal{S}}^{\mathrm{T}}\right)$.

Interestingly, when $\lambda = 0$ and $t = 0$, minimizing $J_{0,0}(\mathcal{S})$ is equivalent to maximizing the trace of the Nystrom approximation of $\mathbf{K}$. Another case for which Eq. (4) simplifies is $t = \lambda$ (this equation was already derived in Yu et al. (2006)):

$$J_{\lambda,\lambda}(\mathcal{S}) = \mathbf{Tr}\left(-\mathbf{K}_{:,\mathcal{S}}\left(\mathbf{K}_{\mathcal{S},\mathcal{S}} + \lambda\mathbf{I}_{|\mathcal{S}|}\right)^{-1}\mathbf{K}_{:,\mathcal{S}}^{\mathrm{T}}\right).$$

Eq. (4) allows us, via the kernel trick, to perform experimental design for learning of nonlinear models defined using high dimensional feature maps. Denote our unlabeled pool of data by $\mathbf{z}_1, \ldots, \mathbf{z}_m \in \mathbb{R}^D$, and that we are using a feature map $\phi : \mathbb{R}^d \to \mathcal{H}$ where $\mathcal{H}$ is some Hilbert space (e.g., $\mathcal{H} = \mathbb{R}^d$), i.e. the regression function is $y(\mathbf{z}) = \langle \phi(\mathbf{z}), \mathbf{w} \rangle_{\mathcal{H}}$. We can then envision the pool of data to be defined by $\mathbf{x}_j = \phi(\mathbf{z}_j)$, $j = 1, \ldots, m$. If we assume we have a kernel function $k : \mathbb{R}^D \times \mathbb{R}^D \to \mathbb{R}^D$ such that $k(\mathbf{x}, \mathbf{z}) = \langle \phi(\mathbf{x}), \phi(\mathbf{z}) \rangle_{\mathcal{H}}$ then $J_{\lambda,t}(\mathcal{S})$ can be computed without actually forming $\mathbf{x}_1, \ldots, \mathbf{x}_m$ since entries in $\mathbf{K}$ can be computed via $k$. If $\mathcal{H}$ is the Reproducing Kernel Hilbert Space of $k$ then this is exactly the setting that corresponds to kernel ridge regression (possibly with a zero ridge term).

**Greedy Algorithm.** We now propose our algorithm for overparameterized experimental design, which is based on greedy minimization of $J_{\lambda,t}(\mathcal{S})$. Greedy algorithms have already been shown to be effective for classical experimental design (Yu et al., 2006; Avron & Boutsidis, 2013; Chamon & Ribeiro, 2017), and it is reasonable to assume this carries on to the overparameterized case.

Our greedy algorithm proceeds as follows. We start with $\mathcal{S}^{(0)} = \emptyset$, and proceed in iteration. At iteration $j$, given selected samples $\mathcal{S}^{(j-1)} \subset [m]$ the greedy algorithm finds the index $i^{(j)} \in [m] - \mathcal{S}^{(j-1)}$ that minimizes $J_{\lambda,t}\left(\mathcal{S}^{(j-1)} \cup \{i^{(j)}\}\right)$. We set $\mathcal{S}^{(j)} \leftarrow \mathcal{S}^{(j-1)} \cup \{i^{(j)}\}$. We continue iterating until $\mathcal{S}^{(j)}$ reaches its target size and/or $J_{\lambda,t}(\mathcal{S})$ is small enough.

The cost of iteration $j$ in a naive implementation is $O\left((m-j)\left(mj^2+j^3\right)\right)$. Through careful matrix algebra, the cost of iteration $j$ can be reduced to $O((m-j)(mj+j^2)) = O(m^2j)$ (since $j \leq m$). The cost of finding a design of size $n$ is then $O(m^2(n^2+D))$ assuming the entire kernel matrix $\mathbf{K}$ is formed at the start and a single evaluation of $k$ takes $O(D)$. Details are delegated to Appendix C.

# 6    SINGLE SHOT DEEP ACTIVE LEARNING

There are few ways in which our proposed experimental design algorithm can be used in the context of deep learning. For example, one can consider a sequential setting where current labeled data are used to create a linear approximation via the Fisher information matrix at the point of minimum loss (Sourati et al., 2018). However, such a strategy falls under the heading of Sequential Experimental Design, and, as we previously stated, in this paper we focus on single shot active learning, i.e. no labeled data is given neither before acquisition nor during acquisition (Yang & Loog, 2019).

In order to design an algorithm for deep active learning, we leverage a recent breakthrough in theoretical analysis of deep learning - the Neural Tangent Kernel (NTK) (Jacot et al., 2018; Lee et al., 2019; Arora et al., 2019a). A rigorous exposition of the NTK is beyond the scope of this paper, but a short and heuristic explanation is sufficient for our needs.

Consider a DNN, and suppose the weights of the various layers can be represented in a vector $\boldsymbol{\theta} \in \mathbb{R}^d$. Given a specific $\boldsymbol{\theta}$, let $f_{\boldsymbol{\theta}}(\cdot)$ denote the function instantiated by that network when the weights are set to $\boldsymbol{\theta}$. The crucial observation is that when the network is wide (width in convolutional layers refers to the number of output channels) enough, we use a quadratic loss function (i.e., $l(f_{\boldsymbol{\theta}}(\mathbf{x}), y) = 1/2(f_{\boldsymbol{\theta}}(\mathbf{x}) - y)^2$), and the initial weights $\boldsymbol{\theta}_0$ are initialized randomly in a standard way, then when training the DNN using gradient descent, the vector of parameters $\boldsymbol{\theta}$ stays almost fixed. Thus, when we consider $\boldsymbol{\theta}_1, \boldsymbol{\theta}_2, \ldots$ formed by training, a first-order Taylor approximation is:

$$f_{\boldsymbol{\theta}_k}(\mathbf{x}) \approx f_{\boldsymbol{\theta}_0}(\mathbf{x}) + \nabla_{\boldsymbol{\theta}} f_{\boldsymbol{\theta}_0}(\mathbf{x})^{\mathrm{T}}(\boldsymbol{\theta}_k - \boldsymbol{\theta}_0)$$

Informally speaking, the approximation becomes an equality in the infinite width limit. The Taylor approximation implies that if we further assume that $\boldsymbol{\theta}_0$ is such that $f_{\boldsymbol{\theta}_0}(\mathbf{x}) = 0$, the learned prediction function of the DNN is well approximated by the solution of a kernel regression problem with the (Finite) Neural Tangent Kernel, defined as

$$k_{f,\boldsymbol{\theta}_0}(\mathbf{x}, \mathbf{z}) := \nabla_{\boldsymbol{\theta}} f_{\boldsymbol{\theta}_0}(\mathbf{x})^{\mathrm{T}} \nabla_{\boldsymbol{\theta}} f_{\boldsymbol{\theta}_0}(\mathbf{z})$$

We remark that there are few simple tricks to fulfill the requirement that $f_{\boldsymbol{\theta}_0}(\mathbf{x}) = 0$.

It has also been shown that under certain initialization distribution, when the width goes to infinity, the NTK $k_{f,\boldsymbol{\theta}_0}$ converges in probability to a deterministic kernel $k_f$ - the *infinite NTK*. Thus, in a sense, instead of training a DNN on a finite width network, we can take the width to infinity and solve a kernel regression problem instead.

Although, it is unclear whether the infinite NTK can be an effective alternative to DNNs in the context of inference, one can postulate that it can be used for deep active learning. That is, in order to select examples to be labeled, use an experimental design algorithm for kernel learning applied to the corresponding NTK. Specifically, for single shot deep active learning, we propose to apply the algorithm presented in the previous section to the infinite NTK. In the next section we present preliminary experiments with this algorithm. We leave theoretical analysis to future research.

# 7    EMPIRICAL EVALUATION

**Transductive vs $\bar{\psi}_{\lambda,0}$ Criterion (i.e., variance-oriented vs. bias-oriented designs)**   $\bar{\psi}_{\lambda,0}$ and $\bar{\psi}_{\lambda,\lambda}$ are simplified version of $\bar{\psi}_{\lambda,t}$ criterion. Our conjecture is that in the overparameterized regime $\bar{\psi}_{\lambda,0}$ is preferable, at least for relatively large $\lambda$. Table 1. empirically supports our conjecture. In this experiment, we performed an experimental design task on 112 classification datasets from UCI database (similar to the list that was used by Arora et al. (2019b) ). Learning is performed using kernel ridge regression with standard RBF kernel. We tried different values of $\lambda$ and checked which criterion brings to a smaller classification error on a test set when selecting 50 samples. Each entry in Table 1 counts how many times $\bar{\psi}_{\lambda,\lambda}$ , won $\bar{\psi}_{\lambda,0}$ won or the error was the same. We consider an equal error when the difference is less the 5%.

Table 1: $\bar{\psi}_{\lambda,0}$ vs $\bar{\psi}_{\lambda,\lambda}$ on UCI datasets. We generated designs on 112 classification datasets. Each cell details the number of datasets in which that selection of $t$ was clearly superior to the other possible choice, or the same (for the "SAME" column).

| $\lambda$ | $\bar{\psi}_{\lambda,\lambda}$ is better | $\bar{\psi}_{\lambda,0}$ is better | SAME |
|---|---|---|---|
| 0.001 | 5 | 8 | 99 |
| 0.01 | 7 | 9 | 96 |
| 0.1 | 16 | 16 | 80 |
| 1.0 | 21 | 43 | 48 |
| 10.0 | 19 | 68 | 25 |

**Deep Active Learning**   Here we report preliminary experiments with the proposed algorithm for single shot deep active learning (Section 6). Additional experiments are reported in the appendix. We used the MNIST dataset, and used the square loss for training. As for the network architecture, we used a version of LeNet5 (LeCun et al., 1998) that is widen by a factor of 8. we refer to this network as "Wide-LeNet5".

The setup is as follows. We use Google's open source neural tangents library (Novak et al., 2020) to compute Gram matrix of the infinite NTK using 59,940 training samples (we did not use the full 60,000 training samples due to batching related technical issues). We then used the algorithm proposed in Section 5 to incrementally select greedy designs of up to 800 samples, where we set the parameters to $\lambda = t = 0$. We now trained the original neural network with different design sizes, each design with five different random initial parameters. Learning was conducted using SGD, with fixed learning rate of 0.1, batch size of 128, and no weight decay. Instead of counting epochs, we simply capped the number of SGD iterations to be equivalent to 20 epochs of the full trainning set. We computed the accuracy of the model predictions on 9963 test-set samples (again, due to technical issues related to batching).

Figure 2 report the mean and standard deviation (over the parameters initialization) of the final accuracy. We see a consistent advantage in terms of accuracy for designs selected via our algorithm, though as expected the advantage shrinks as the training size increase. Notice, that comparing the accuracy of our design with 400 training samples, random selection required as many as 600 for Wide-LeNet5 to achieve the same accuracy!

Two remarks are in order. First, to prevent overfitting and reduce computational load, at each iteration of the greedy algorithm we computed the score for only on a subset of 2000 samples from the pool. Second, to keep the experiment simple we refrained from using tricks that ensure $f_{\boldsymbol{\theta}_0} = 0$.

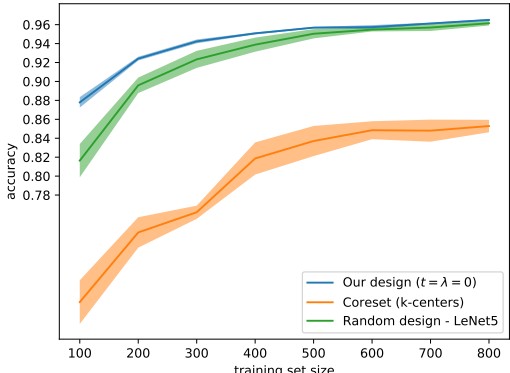

Figure 2: Single shot active learning with Wide-LeNet5 model on MNIST.

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

## A PROOFS

### A.1 PROOF OF PROPOSITION 1

*Proof.* We prove the case of $d \geq n$ (for $\mathbf{X} \in \mathbb{R}^{n \times d}$). The proof for $d < n$ is similar. It is enough to show that $\lim_{\lambda \to 0} \mathbf{X}_\lambda^+ = \mathbf{X}^+$. For a scalar $\gamma$ let

$$\gamma^+ := \begin{cases} \gamma^{-1} & \gamma \neq 0 \\ 0 & \gamma = 0 \end{cases}$$

Let $\mathbf{X} = \mathbf{U}\Sigma\mathbf{V}^{\mathrm{T}}$ be the SVD of $\mathbf{X}$ with

$$\Sigma = \begin{bmatrix} \sigma_1 & & & 0 & \cdots & 0 \\ & \ddots & & \vdots & & \vdots \\ & & \sigma_d & 0 & \cdots & 0 \end{bmatrix} \in \mathbb{R}^{d \times n}$$

where $\sigma_1, \ldots, \sigma_d$ are the singular values of $\mathbf{X}$. We have $\mathbf{X}^+ = \mathbf{V}\Sigma^+\mathbf{U}^{\mathrm{T}}$ where we have

$$\Sigma^+ = \begin{bmatrix} \sigma_1^+ & & \\ & \ddots & \\ & & \sigma_d^+ \\ 0 & \cdots & 0 \\ \vdots & & \vdots \\ 0 & \cdots & 0 \end{bmatrix}$$

On the other hand, simple matrix algebra shows that

$$\mathbf{X}_\lambda^+ = \mathbf{V} \begin{bmatrix} (\sigma_1^2 + \lambda)^+ \sigma_1 & & \\ & \ddots & \\ & & (\sigma_d^2 + \lambda)^+ \sigma_d \\ 0 & \cdots & 0 \\ \vdots & & \vdots \\ 0 & \cdots & 0 \end{bmatrix} \mathbf{U}^{\mathrm{T}} \tag{5}$$

Now clearly for $i = 1, \ldots, d$,

$$\lim_{\lambda \to 0^+} (\sigma_i^2 + \lambda)^+ \sigma_i = \sigma_i^+$$

So the limit of the diagonal matrix in Eq. (5) when $\lambda \to 0^+$ is $\Sigma^+$. Since matrix product is a linear, and thus continuous function, the proposition follows. □

## A.2 Proof of Proposition 2

*Proof.* Let us write

$$\epsilon := \begin{bmatrix} \epsilon_1 \\ \vdots \\ \epsilon_n \end{bmatrix}$$

so

$$\mathbf{y} = \mathbf{X}\mathbf{w} + \epsilon \,.$$

Thus,

$$\hat{\mathbf{w}}_\lambda = \mathbf{X}_\lambda^+ \mathbf{y} = \mathbf{X}_\lambda^+ \mathbf{X}\mathbf{w} + \mathbf{X}_\lambda^+ \epsilon = \mathbf{M}_\lambda^+ \mathbf{M}\mathbf{w} + \mathbf{X}_\lambda^+ \epsilon$$

and

$$\mathbf{x}^\mathsf{T}\mathbf{w} - \mathbf{x}^\mathsf{T}\hat{\mathbf{w}}_\lambda = \mathbf{x}^\mathsf{T}(\mathbf{I}_d - \mathbf{M}_\lambda^+ \mathbf{M})\mathbf{w} + \mathbf{x}^\mathsf{T}\mathbf{X}_\lambda^+ \epsilon$$

For brevity we denote $\mathbf{P}_{\perp\mathbf{X}}^\lambda = \mathbf{I}_d - \mathbf{M}_\lambda^+ \mathbf{M}$. Note that this is not really a projection, but rather (informally) a "soft projection". So:

$$(\mathbf{x}^\mathsf{T}\mathbf{w} - \mathbf{x}^\mathsf{T}\hat{\mathbf{w}}_\lambda)^2 = \mathbf{w}^\mathsf{T}\mathbf{P}_{\perp\mathbf{X}}^\lambda(\mathbf{x}\mathbf{x}^\mathsf{T})\mathbf{P}_{\perp\mathbf{X}}^\lambda\mathbf{w} + \mathbf{w}^\mathsf{T}\mathbf{P}_{\perp\mathbf{X}}^\lambda(\mathbf{x}\mathbf{x}^\mathsf{T})\mathbf{X}_\lambda^+ \epsilon + \epsilon^\mathsf{T}(\mathbf{X}_\lambda^+)^\mathsf{T}(\mathbf{x}\mathbf{x}^\mathsf{T})\mathbf{X}_\lambda^+ \epsilon$$

Finally,

$$
\begin{aligned}
\mathbb{E}\left[R(\hat{\mathbf{w}}_\lambda)\right] &= \mathbb{E}_{\mathbf{x},\epsilon}\left[\left(\mathbf{x}^\mathsf{T}\mathbf{w} - \mathbf{x}^\mathsf{T}\hat{\mathbf{w}}_\lambda\right)^2\right] \\
&= \mathbb{E}_\epsilon\left[\mathbb{E}_{\mathbf{x}}\left[\left(\mathbf{x}^\mathsf{T}\mathbf{w} - \mathbf{x}^\mathsf{T}\hat{\mathbf{w}}_\lambda\right)^2 \mid \epsilon\right]\right] \\
&= \mathbb{E}_\epsilon\left[\mathbb{E}_{\mathbf{x}}\left[\mathbf{w}^\mathsf{T}\mathbf{P}_{\perp\mathbf{X}}^\lambda(\mathbf{x}\mathbf{x}^\mathsf{T})\mathbf{P}_{\perp\mathbf{X}}^\lambda\mathbf{w} + \mathbf{w}^\mathsf{T}\mathbf{P}_{\perp\mathbf{X}}^\lambda(\mathbf{x}\mathbf{x}^\mathsf{T})\mathbf{X}_\lambda^+ \epsilon + \epsilon^\mathsf{T}(\mathbf{X}_\lambda^+)^\mathsf{T}(\mathbf{x}\mathbf{x}^\mathsf{T})\mathbf{X}_\lambda^+ \epsilon \mid \epsilon\right]\right] \\
&= \mathbb{E}_\epsilon\left[\mathbf{w}^\mathsf{T}\mathbf{P}_{\perp\mathbf{X}}^\lambda\mathbf{C}_\rho\mathbf{P}_{\perp\mathbf{X}}^\lambda\mathbf{w} + \mathbf{w}^\mathsf{T}\mathbf{P}_{\perp\mathbf{X}}^\lambda\mathbf{C}_\rho\mathbf{X}_\lambda^+ \epsilon + \epsilon^\mathsf{T}(\mathbf{X}_\lambda^+)^\mathsf{T}\mathbf{C}_\rho\mathbf{X}_\lambda^+ \epsilon\right] \\
&= \mathbf{w}^\mathsf{T}\mathbf{P}_{\perp\mathbf{X}}^\lambda\mathbf{C}_\rho\mathbf{P}_{\perp\mathbf{X}}^\lambda\mathbf{w} + \sigma^2\mathbf{Tr}\left((\mathbf{X}_\lambda^+)^\mathsf{T}\mathbf{C}_\rho\mathbf{X}_\lambda^+\right) \\
&= \|\mathbf{C}_\rho^{1/2}\mathbf{P}_{\perp\mathbf{X}}^\lambda\mathbf{w}\|_2^2 + \sigma^2\mathbf{Tr}\left(\mathbf{C}_\rho\mathbf{X}_\lambda^+(\mathbf{X}_\lambda^+)^\mathsf{T}\right) \\
&= \|\mathbf{C}_\rho^{1/2}\left(\mathbf{I} - \mathbf{M}_\lambda^+ \mathbf{M}\right)\mathbf{w}\|_2^2 + \sigma^2\mathbf{Tr}\left(\mathbf{C}_\rho\mathbf{M}_\lambda^{+^2}\mathbf{M}\right)
\end{aligned}
$$

$\square$

## A.3 Proof of Proposition 3

Before proving Proposition 3 we need the following definition and theorem.

**Definition 1.** For a family of sets $\{A_\lambda\}_{\lambda\in\mathbb{R}}$, $A \subset \mathbb{R}^d$ we write $\lim_{\lambda\to\bar{\lambda}} A_\lambda = A$ if $\mathbf{w} \in A$ if and only if there exists sequence $\lambda_n \to \lambda$ and a sequence $\mathbf{w}_n \to \mathbf{w}$ where $\mathbf{w}_n \in A_{\lambda_n}$ for sufficiently large $n$.

**Theorem 1.** *(A restricted version of Theorem 1.17 in Rockafellar & Wets (2009)) Consider $f$ : $\Omega \times \Psi \to \mathbb{R}$ where $\Omega \subseteq \mathbb{R}^d$ and $\Psi \subseteq \mathbb{R}$ are compact and $f$ is continuous. Then*

$$\lim_{\lambda\to\bar{\lambda}} \operatorname*{argmin}_{\mathbf{w}} f(\mathbf{w}, \lambda) \subseteq \operatorname*{argmin}_{\mathbf{w}} f(\mathbf{w}, \bar{\lambda}) \,.$$

*Proof.* Suppose $\bar{\mathbf{w}} \in \lim_{\lambda\to\bar{\lambda}} \operatorname{argmin}_{\mathbf{w}} f(\mathbf{w}, \lambda)$. The implies that there exits $\lambda_n \to \bar{\lambda}$ such that $\mathbf{w}_n \in \operatorname{argmin}_{\mathbf{w}} f(\mathbf{w}, \lambda_n)$ and $\mathbf{w}_n \to \bar{\mathbf{w}}$. From the continuity of $f$ we have that $f(\mathbf{w}_n, \lambda_n) \to f(\bar{\mathbf{w}}, \bar{\lambda})$. Now suppose for the sake of contradiction that $\bar{\mathbf{w}} \notin \operatorname{argmin}_{\mathbf{w}} f(\mathbf{w}, \bar{\lambda})$. So there is $\mathbf{u}$ such that $f(\mathbf{u}, \bar{\lambda}) < f(\bar{\mathbf{w}}, \bar{\lambda})$. From the continuity of $f$ in $\lambda$ there is $n_0$ such that for all $n > n_0$ $f(\mathbf{u}, \lambda_n) < f(\bar{\mathbf{w}}, \bar{\lambda})$. Then from the continuity of $f$ in $\mathbf{w}$, and $\mathbf{w}_n \to \bar{\mathbf{w}}$, for sufficiently large $n$, $f(\mathbf{w}_n, \lambda_n) > f(\mathbf{u}, \lambda_n)$, which contradicts $\mathbf{w}_n \in \operatorname{argmin}_{\mathbf{w}} f(\mathbf{w}, \lambda_n)$.

$\square$

We are now ready to prove Proposition 3.

*Proof.* Consider the function $f(\mathbf{M}, \mu) = \mathbf{Tr}\left(\mathbf{C}_\rho \mathbf{M}_\mu^{-1}\right)$

$$f(\mathbf{M}, \mu) = \begin{cases} \mathbf{Tr}\left(\mu\mathbf{C}_\rho(\mathbf{M} + \mu\mathbf{I})^{-1}\right) & \mu > 0 \\ \mathbf{Tr}\left(\mathbf{C}_\rho\left(\mathbf{I} - \mathbf{M}^+\mathbf{M}\right)\right) & \mu = 0 \end{cases}$$

defined over $\Omega \times \mathbb{R}_{\geq 0}$ where $\mathbb{R}_{\geq 0}$ denotes the set of non-negative real numbers. Note that this function is well-defined since $\Omega$ is a set of positive semidefinite matrices.

We now show that $f$ is continuous. For $\mu > 0$ it is clearly continuous for every $\mathbf{M}$, so we focus on the case that $\mu = 0$ for an arbitrary $\mathbf{M}$. Consider a sequence $\mathbb{R}_{>0} \ni \mu_n \to 0$ (where $\mathbb{R}_{>0}$ is the set of positive reals) and $\Omega \ni \mathbf{M}_n \to \mathbf{M}$. Since $\Omega$ is compact, $\mathbf{M} \in \Omega$. Let us write a spectral decomposition of $\mathbf{M}_n$ (recall that $\Omega$ is a set of symmetric matrices)

$$\mathbf{M}_n = \mathbf{U}_n \Lambda_n \mathbf{U}_n^{\mathsf{T}}$$

where $\Lambda_n$ is diagonal with non-negative diagonal elements (recall that $\Omega$ is a set of positive definite matrices). Let $\mathbf{M} = \mathbf{U}\Lambda\mathbf{U}^{\mathsf{T}}$ be a spectral decomposition of $\mathbf{M}$. Without loss of generality we may assume that $\mathbf{U}_n \to \mathbf{U}$ and $\Lambda_n \to \Lambda$. Now note that

$$(\mathbf{M}_n + \mu_n\mathbf{I})^{-1}\mathbf{M}_n = \mathbf{U}_n(\Lambda_n + \mu_n\mathbf{I})^{-1}\Lambda_n\mathbf{U}_n^{\mathsf{T}}$$

One can easily show that $(\Lambda_n + \mu_n\mathbf{I})^{-1}\Lambda_n \to \text{sign}(\Lambda)$ where sign is taken entry wise, which implies that $(\mathbf{M}_n + \mu_n\mathbf{I})^{-1}\mathbf{M}_n \to \mathbf{U}\,\text{sign}(\Lambda)\mathbf{U}^{\mathsf{T}}$ since matrix multiplication is continuous. Next, note that $\mathbf{M}^+\mathbf{M} = \mathbf{U}\Lambda^+\Lambda\mathbf{U}^{\mathsf{T}} = \mathbf{U}\,\text{sign}(\Lambda)\mathbf{U}^{\mathsf{T}}$ so $(\mathbf{M}_n + \mu_n\mathbf{I})^{-1}\mathbf{M}_n \to \mathbf{M}^+\mathbf{M}$. The Woodbury formula implies that

$$\mu_n\mathbf{C}_\rho(\mathbf{M}_n + \mu_n\mathbf{I})^{-1} = \mathbf{C}_\rho\left(\mathbf{I} - (\mathbf{M}_n + \mu_n\mathbf{I})^{-1}\mathbf{M}_n\right)$$

so the continuity of the trace operator implies that

$$\mathbf{Tr}\left(\mu_n\mathbf{C}_\rho(\mathbf{M}_n + \mu_n\mathbf{I})^{-1}\right) = \mathbf{Tr}\left(\mathbf{C}_\rho\left(\mathbf{I} - (\mathbf{M}_n + \mu_n\mathbf{I})^{-1}\mathbf{M}_n\right)\right) \to \mathbf{Tr}\left(\mathbf{C}_\rho\left(\mathbf{I} - \mathbf{M}^+\mathbf{M}\right)\right)$$

which shows that $f$ is continuous.

Theorem 1 now implies the claim since for $\mu > 0$ we have

$$\underset{\mathbf{M}\in\Omega}{\arg\min}\,\mathbf{Tr}\left(\mathbf{C}_\rho\mathbf{M}_\mu^{-1}\right) = \underset{\mathbf{M}\in\Omega}{\arg\min}\,\mathbf{Tr}\left(\mu\mathbf{C}_\rho\mathbf{M}_\mu^{-1}\right).$$

$\square$

### A.4 PROOF OF PROPOSITION 4

*Proof.* Let $\mathbf{A} = \left(\mathbf{I} - (\mathbf{M} + \lambda\mathbf{I}_d)^{-1}\mathbf{M}\right)^2 + \lambda(\mathbf{M} + \lambda\mathbf{I}_d)^{-2}\mathbf{M}$, so $\bar{\varphi}_{\lambda,\lambda}(\mathbf{M}) = \mathbf{Tr}\left(\mathbf{C}_\rho\mathbf{A}\right)$. We now have for $\lambda > 0$:

$$\begin{aligned} \mathbf{A} &= \left(\mathbf{I} - (\mathbf{M} + \lambda\mathbf{I}_d)^{-1}(\mathbf{M} + \lambda\mathbf{I}_d) + \lambda(\mathbf{M} + \lambda\mathbf{I}_d)^{-1}\right)^2 + \lambda(\mathbf{M} + \lambda\mathbf{I}_d)^{-2}\mathbf{M} \\ &= \lambda^2\left(\mathbf{M} + \lambda\mathbf{I}_d\right)^{-2} + \lambda(\mathbf{M} + \lambda\mathbf{I}_d)^{-2}\mathbf{M} \\ &= \lambda(\mathbf{M} + \lambda\mathbf{I}_d)^{-2}\left(\mathbf{M} + \lambda\mathbf{I}_d\right) \\ &= \lambda\left(\mathbf{M} + \lambda\mathbf{I}\right)^{-1} \end{aligned}$$

so:

$$\mathbf{Tr}\left(\mathbf{C}_\rho\mathbf{A}\right) = \lambda\mathbf{Tr}\left(\mathbf{C}_\rho\left(\mathbf{M} + \lambda\mathbf{I}\right)^{-1}\right).$$

Since $\lambda > 0$ it doesn't affect the minimizer. $\square$

## B RELATION TO CORESETS

The idea in the coreset approach for active learning is to find an $\mathcal{S}$ such that

$$C(\mathcal{S}) = \left| \frac{1}{m}\sum_{i=1}^{m} l(\mathbf{x}_i, y_i \mid \mathcal{S}) - \frac{1}{|\mathcal{S}|}\sum_{i\in\mathcal{S}} l(\mathbf{x}_i, y_i \mid \mathcal{S}) \right|$$

is minimized. In the above $l(\mathbf{x}, y \mid \mathcal{S})$ is a loss function, and the conditioning on $\mathcal{S}$ denotes that the parameters of the loss function are the ones obtained when training only using indices selected in $\mathcal{S}$. For linear regression the conditioning on $\mathcal{S}$ is not relevant (since the parameters do not affect the loss). The motivation for minimizing $C(\mathcal{S})$ is that the expected test loss can be broken to the generalization loss on the entire dataset (which is fixed), the training loss (which is 0 in the presence of overparameterization) and the coreset loss.

One popular approach to active learning using coresets is to find a *coverset*. A $\delta$-coverset of a set of points $\mathcal{A}$ is a set of points $\mathcal{B}$ such that for every $\mathbf{x} \in \mathcal{A}$ there exists a $\mathbf{y} \in \mathcal{B}$ such that $\|\mathbf{x} - \mathbf{y}\|_2 \leq \delta$ (other metrics can be used as well). Sener and Savarese Sener & Savarese (2018) showed that under suitable Lipschitz and boundness conditions, if $\{\mathbf{x}_i\}_{i \in \mathcal{S}}$ is a $\delta$-coverset of $\{\mathbf{x}_i\}_{i \in [m]}$ then

$$C(\mathcal{S}) \leq O(\delta + m^{-1/2})$$

which motivates finding a $\mathcal{S}$ that minimizes $\delta_{\mathcal{S}}$, where $\delta_{\mathcal{S}}$ denotes the minimal $\delta$ for which $\{\mathbf{x}_i\}_{i \in \mathcal{S}}$ is a $\delta$-coverset of $\{\mathbf{x}_i\}_{i \in [m]}$.

Since for a $\mathbf{x}$ in the training set (which is a row of $\mathbf{V}$) $\|\mathbf{x}(\mathbf{I}_d - \mathbf{P_M})\|_2^2$, for $\mathbf{M} = \mathbf{V}_{\mathcal{S},:}^{\mathsf{T}} \mathbf{V}_{\mathcal{S},:}$ is the minimal distance from $\mathbf{x}$ to the span of $\{\mathbf{x}_i\}_{i \in \mathcal{S}}$, and as such is always smaller than the distance between $\mathbf{x}$ and it's closest point in $\{\mathbf{x}_i\}_{i \in \mathcal{S}}$, it is easy to show that

$$n^{-1} \bar{\psi}_{0,0}(\mathbf{V}_{\mathcal{S},:}^{\mathsf{T}} \mathbf{V}_{\mathcal{S},:}) \leq \delta_{\mathcal{S}}^2.$$

Thus, minimizing $\delta_{\mathcal{S}}$ can be viewed as minimizing an upper bound on the bias term when $\lambda = 0$.

Under the setup of the experiment in Section 7 we tried to replace our design with *k-centers* algorithm, which often used as approximated solution for the problem of finding $\mathcal{S}$ that minimizes $\delta_{\mathcal{S}}$. How ever the result we got were much worse then random design, probably due to the problem of outliers. We did not try more sophisticated versions of the k-center algorithm that tackle the problem of outliers.

## C   DETAILS ON THE ALGORITHM

We discuss the case of $\lambda = 0$. The case of $\lambda > 0$ requires some more careful matrix algebra, so we omit the details.

Let us define

$$\mathbf{A}_j := \mathbf{K}_{\mathcal{S}^{(j)}, \mathcal{S}^{(j)}}^{-1}, \quad \mathbf{B}_j := \mathbf{K}_{:, \mathcal{S}^{(j)}}^{\mathsf{T}} \mathbf{K}_{:, \mathcal{S}^{(j)}}$$

and note that $J_{\lambda,t}(\mathcal{S}^{(j)}) = -\mathbf{Tr}\left(\mathbf{B}_j(\mathbf{A}_j - t\mathbf{A}_j^2)\right)$. We also denote by $\tilde{\mathbf{A}}_j$ and $\tilde{\mathbf{B}}_j$ the matrices obtained from $\mathbf{A}_j$ and $\mathbf{B}_j$ (respectively) by adding a zero row and column.

Our goal is to efficiently compute $J_{\lambda,t}(\mathcal{S}^{(j-1)} \cup \{i\})$ for any $i \in [m] - \mathcal{S}^{(j-1)}$ so find $i^{(j)}$ and form $\mathcal{S}^{(j)}$. We assume that at the start of iteration $j$ we already have in memory $\mathbf{A}_{j-1}$ and $\mathbf{B}_{j-1}$. We show later how to efficiently update $\mathbf{A}_j$ and $\mathbf{B}_j$ once we have found $i^{(j)}$. For brevity, let us denote

$$\mathcal{S}_i^{(j)} := \mathcal{S}^{(j-1)} \cup \{i\}, \quad \mathbf{A}_{ji} := \mathbf{K}_{\mathcal{S}_i^{(j)}, \mathcal{S}_i^{(j)}}^{-1}, \quad \mathbf{B}_{ji} := \mathbf{K}_{:, \mathcal{S}_i^{(j)}}^{\mathsf{T}} \mathbf{K}_{:, \mathcal{S}_i^{(j)}}$$

Let us also define

$$\mathbf{C}_{j-1} := \tilde{\mathbf{B}}_{j-1} \tilde{\mathbf{A}}_{j-1}, \quad \mathbf{D}_{j-1} := \tilde{\mathbf{B}}_{j-1} \tilde{\mathbf{A}}_{j-1}^2, \quad \mathbf{E}_{j-1} := \tilde{\mathbf{A}}_{j-1}^2$$

Again, we assume that at the start of iteration $j$ we already have in memory $\mathbf{C}_{j-1}$, $\mathbf{D}_{j-1}$ and $\mathbf{E}_{j-1}$, and show how to efficiently update these.

Let

$$\mathbf{W}_{ji} := \left[ \begin{array}{cc} 0_{j-1} & \mathbf{K}_{:, \mathcal{S}^{(j-1)}}^{\mathsf{T}} \mathbf{K}_{:, i} \\ \mathbf{K}_{:, i}^{\mathsf{T}} \mathbf{K}_{:, \mathcal{S}^{(j-1)}} & \mathbf{K}_{:, i}^{\mathsf{T}} \mathbf{K}_{:, i} \end{array} \right]$$

and note that

$$\mathbf{B}_{ji} = \tilde{\mathbf{B}}_{j-1} + \mathbf{W}_{ji}.$$

Also important is the fact that $\mathbf{W}_{ji}$ has rank 2 and that finding the factors takes $O(mj)$ discounting the cost of computing columns of $\mathbf{K}$. Next, let us denote

$$r_{ji} = \frac{1}{(\mathbf{K}_{ii} - \mathbf{K}_{\mathcal{S}^{(j)}, i}^{\mathsf{T}} \mathbf{A}_{j-1} \mathbf{K}_{\mathcal{S}^{(j)}, i})}$$

and

$$\mathbf{Q}_{ji} := r_{ji} \cdot \begin{bmatrix} \mathbf{A}_{j-1}\mathbf{K}_{\mathcal{S}^{(j)},i}\mathbf{K}_{\mathcal{S}^{(j)},i}^{\mathrm{T}}\mathbf{A}_{j-1}^{-1} & -\mathbf{A}_{j-1}\mathbf{K}_{\mathcal{S}^{(j)},i} \\ -\mathbf{K}_{\mathcal{S}^{(j)},i}^{\mathrm{T}}\mathbf{A}_{j-1} & 1 \end{bmatrix}$$

A well known identity regarding Schur complement implies that

$$\mathbf{A}_{ji} = \tilde{\mathbf{A}}_{j-1} + \mathbf{Q}_{ji}$$

Also important is the fact that $\mathbf{Q}_{ji}$ has rank 2 and that finding the factors takes $O(j^2)$ discounting the cost of computing entries of $\mathbf{K}$.

So

$$\begin{aligned}
J_{\lambda,t}(\mathcal{S}_i^{(j)}) &= -\mathbf{Tr}\left(\mathbf{B}_{ji}(\mathbf{A}_{ji} - t\mathbf{A}_{ji}^2)\right) \\
&= -\mathbf{Tr}\left((\tilde{\mathbf{B}}_{j-1} + \mathbf{W}_{ji})(\tilde{\mathbf{A}}_{j-1} + \mathbf{Q}_{ji} - t(\tilde{\mathbf{A}}_{j-1} + \mathbf{Q}_{ji})^2)\right) \\
&= -\mathbf{Tr}\left((\tilde{\mathbf{B}}_{j-1} + \mathbf{W}_{ji})(\tilde{\mathbf{A}}_{j-1} + \mathbf{Q}_{ji}) - t(\tilde{\mathbf{B}}_{j-1} + \mathbf{W}_{ji})(\tilde{\mathbf{A}}_{j-1}^2 + \mathbf{Q}_{ji}^2 + \tilde{\mathbf{A}}_{j-1}\mathbf{Q}_{ji} + \mathbf{Q}_{ji}\tilde{\mathbf{A}}_{j-1})\right) \\
&= -\mathbf{Tr}\left(\mathbf{C}_{j-1} + \tilde{\mathbf{B}}_{j-1}\mathbf{Q}_{ji} + \mathbf{W}_{ji}(\tilde{\mathbf{A}}_{j-1} + \mathbf{Q}_{ji})\right) \\
&\quad + t\mathbf{Tr}\left(\mathbf{D}_{j-1} + \tilde{\mathbf{B}}_j(\tilde{\mathbf{A}}_{j-1}\mathbf{Q}_{ji} + \mathbf{Q}_{ji}\tilde{\mathbf{A}}_{j-1} + \mathbf{Q}_{ji}^2)\right) \\
&\quad + \mathbf{Tr}\left(\mathbf{W}_i(\mathbf{E}_{j-1} + \mathbf{Q}_{ji}^2 + \tilde{\mathbf{A}}_{j-1}\mathbf{Q}_{ji} + \mathbf{Q}_{ji}\tilde{\mathbf{A}}_{j-1})\right)
\end{aligned}$$

Now, $\mathbf{C}_{j-1}$ is already in memory so $\mathbf{Tr}\left(\mathbf{C}_{j-1}\right)$ can be computed in $O(j)$, $\mathbf{Q}_{ji}$ has rank 2 and $\tilde{\mathbf{B}}_{j-1}$ is in memory so $\mathbf{Tr}\left(\tilde{\mathbf{B}}_{j-1}\mathbf{Q}_{ji}\right)$ can be compute in $O(j^2)$, and $\mathbf{W}_{ji}$ has rank 2 and $\tilde{\mathbf{A}}_{j-1}$ is in memory so $\mathbf{Tr}\left(\mathbf{W}_i(\tilde{\mathbf{A}}_{j-1} + \mathbf{Q}_{ji})\right)$ can be computed in $O(j^2)$. Using a similar rationale, all the other terms of $J_{\lambda,t}(\mathcal{S}_i^{(j)})$ can also be computed in $O(j)$ or $O(j^2)$, and overall $J_{\lambda,t}(\mathcal{S}_i^{(j)})$ can be computed in $O(j^2)$. Thus, scanning for $i^{(j)}$ takes $O((m-j)j^2)$.

Once $i^{(j)}$ has been identified, we set $\mathcal{S}^{(j)} = \mathcal{S}_{i^{(j)}}^{(j)}$, $\mathbf{A}_j = \mathbf{A}_{ji^{(j)}} = \tilde{\mathbf{A}}_{j-1} + \mathbf{Q}_{ji^{(j)}}$ and $\mathbf{B}_j = \mathbf{B}_{ji^{(j)}} = \tilde{\mathbf{B}}_{j-1} + \mathbf{W}_{ji^{(j)}}$. The last two can be computed in $O(j^2)$ once we form $\mathbf{Q}_{i^{(j)}}$ and $\mathbf{W}_{i^{(j)}}$. Computing the factors of these matrices takes $O(mj)$. As for updating $\mathbf{C}_{j-1}$, we have

$$\mathbf{C}_j = \tilde{\mathbf{C}}_{j-1} + \tilde{\mathbf{B}}_{j-1}\mathbf{Q}_{ji^{(j)}} + \mathbf{W}_{ji^{(j)}}\tilde{\mathbf{A}}_{j-1} + \mathbf{W}_{ji^{(j)}}\mathbf{Q}_{ji^{(j)}}$$

where $\tilde{\mathbf{C}}_{j-1}$ is obtained from $\mathbf{C}_{j-1}$ be adding a zero row and column. Since $\mathbf{C}_{j-1}$ is in memory and both $\mathbf{Q}_{ji^{(j)}}$ and $\mathbf{W}_{i^{(j)}}$ have rank $O(1)$, we can compute $\mathbf{C}_j$ is $O(j^2)$. Similar reasoning can be used to show that $\mathbf{D}_j$ and $\mathbf{E}_j$ can also be computed in $O(j^2)$.

Overall, the cost of iteration $j$ is $O((m-j)(mj + j^2)) = O(m^2j)$ (since $j \le m$). The cost of finding a design of size $n$ is $O(m^2(n^2 + D))$ assuming the entire kernel matrix $\mathbf{K}$ is formed at the start and a single evaluation of $k$ takes $O(D)$.

## D EXPERIMENTAL PARAMETERS EXPLORATION AND COMPARISON TO TRANSDUCTIVE EXPERIMENTAL DESIGN

In this subsection we report a set of experiments on a kernel ridge regression setup (though in one experiment we set the ridge term to 0, so we are using interpolation). We use the MNIST handwriting dataset (LeCun et al., 2010), where the regression target response was computed by applying one-hot function on the labels 0-9. Nevertheless, we still measure the MSE, and do not use the learnt models as classifiers. We use the RBF kernel $k(\mathbf{x}, \mathbf{z}) = \exp(-\gamma\|\mathbf{x} - \mathbf{z}\|_2^2)$ with parameter $\gamma = 1/784$. From the dataset, we used the standard test set of 10000 images and selected randomly another 10000 images from the rest of the 60000 images as a pool. We used our proposed greedy algorithm to select a training set of sizes 1 to 100. We use two values of $\lambda$: $\lambda = 0$ (interpolation), and $\lambda = 0.75^2$. The optimal $\lambda$ according to cross validation was the smallest we checked so we just used $\lambda = 0$. However, in some cases having a $\lambda > 0$ is desirable from a computational perspective, e.g. it caps the condition

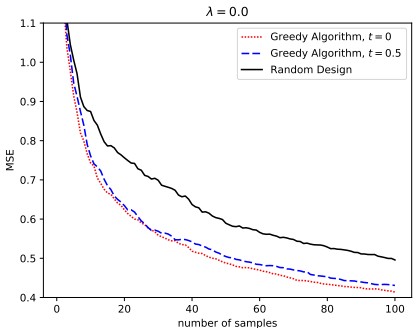 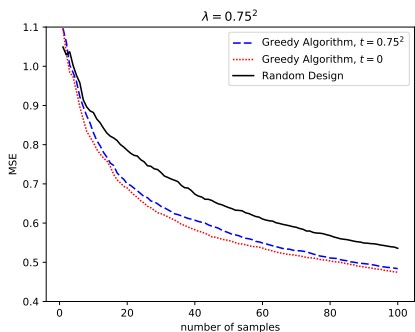

Figure 3: Kernel regression experiments on MNIST.

number of the kernel matrix, making the linear system easier to solve. Furthermore, in real world scenarios, oftentimes we do not have any data before we start to acquire labels, and if we do, it is not always distributed as in the test data, so computing the optimal $\lambda$ can be a challenging.

Results are reported in Figure 3. The left panel show the results for $\lambda = 0$. We report results for $t = 0$ and $t = 0.5$. The choice of $t = 0$ worked better. Kernel models with the RBF kernel are highly overparameterized (the hypothesis space is infinite dimensional), so we expect the MSE to be bias dominated, in which case a small $t$ (or $t = 0$) might work best. Recall that the option of $\lambda = t = 0$ is equivalent to the Column Subset Selection Problem, is the limit case of transductive experimental design (Yu et al., 2006), and can be related to the coreset approach (specifically Sener & Savarese (2018)).

The case of $\lambda = 0.75^2$ is reported in the right panel of Figure 3. We tried $t = 0$ and $t = \lambda = 0.75^2$. Here too, using a purely bias oriented objective (i.e., $t = 0$) worked better. Note that this is in contrast with classical OED which use variance oriented objectives. The choice of $t = \lambda$ worked well, but not optimally. In general, in the reported experiments, and other experiments conducted but not reported, it seems that the choice of $t = \lambda$, which is, as we have shown in this paper, equivalent to transductive experimental design, usually works well, but is not optimal.

## E  EXPERIMENTAL SETUP FOR RESULT REPORTED IN FIGURE 1

First, $\mathbf{w} \in \mathbb{R}^{100}$ was sampled randomly from $\mathcal{N}(0, \mathbf{I})$ . Then a pool (the set from which we later choose the design) of 500 samples and a test set of 100 samples were randomly generated according to $\mathbf{x} \sim \mathcal{N}(0, \Sigma)$, $\epsilon \sim \mathcal{N}(0, \sigma^2 \mathbf{I})$ and $y = \mathbf{x}^\mathsf{T}\mathbf{w} + \epsilon$, where $\Sigma \in \mathbb{R}^{100 \times 100}$ is diagonal with $\Sigma_{ii} = \exp(-2.5i/100)$, and $\sigma = 0.2$. We then created three incremental designs (training sets) of size 120 according to three different methods:

- Random design - at each iteration we randomly choose the next training sample from the remaining pool.
- Classical OED (variance oriented) - at each iteration we choose the next training sample from the remaining pool with a greedy step that minimizes the variance term in Eq. (3).
- Overparameterized OED - at each iteration we chose the next training sample from the remaining pool with a greedy step that minimizes Eq. (3), with $\lambda = 0$ and $t = \sigma^2$ .

With the addition of each new training sample we computed the new MSE achieved on the test set with minimum norm linear regression.

## F  EXPERIMENT: SINGLE SHOT ACTIVE LEARNING FOR NARROW NETWORKS

In Figure 4 we compare the result of our method on LeNet5 with the result of our method on Wide-LeNet5. We see that while the result on

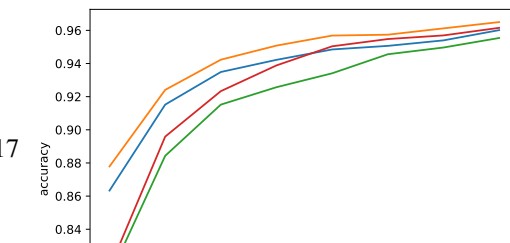

the wide version are generally better, both for random designs and our design, our method brings a consistent advantage over random design. In both the narrow and the wide versions it requires about 600 training samples for the random design to achieve the accuracy achieved using our algorithm with only 400 training samples!

The parameters used by our algorithm to select the design are $\lambda = t = 0$. For the network training we used SGD with batch size 128, leaning rate 0.1 and no regularization. The SGD number iterations is equivalent to 20 epochs of the full trainning set.

# G    SEQUENTIAL VS SINGLE SHOT ACTIVE LEARNING

While in this work focus on the single shot active learning, an interesting question is how does it compare to sequential active learning. In sequential active learning we alternate between a model improving step and a step of new labels acquisition,. This obviously gives an advantage to sequential active learning over single shot active learning, as the latter is a restricted instance of the former.

As we still do not have a sequential version of our algorithm to compare with, we chose to experimentally compare our single shot algorithm with the classical method of *uncertainty sampling* (Bartlett et al., 2020). This method has proved to be relatively efficient for neural networks (Gal et al., 2017). Uncertainty sampling based active learning requires computing the uncertainty of the updated model regarding each sample in the pool. As such, this approach is sequential by nature.

Usually uncertainty sampling is derived in connection to the cross entropy since in that case the network output after the $\mathrm{softmax}$ layer can be interpreted as a probability estimation of $y = i$ given $\mathbf{x}$, which we symbolize as $p_i(\mathbf{x})$. The uncertainty score (in one common version) is then given by

$$1 - \max_{i \in [L]} p_i(\mathbf{x}).$$

Because we use the square lose, we need to make some adaptation for the way of $p_i(\mathbf{x})$ is computed. Considering the fact that the square loss is an outcome of a *maximum likelihood* model that given $\mathbf{x}$ assumes $\mathbf{y} \sim \mathcal{N}(f(\mathbf{x}), \mathbf{I}_L)$, it make sense to use

$$p_i(\mathbf{x}) = (2\pi)^{-\frac{L}{2}} e^{-\frac{1}{2}\|\mathbf{y}_i - f(\mathbf{x})\|_2^2},$$

where $\mathbf{y}_i$ is the $\mathrm{onehot}$ vector of $i$.

Figure 5. shows a comparison between the accuracy achieved with our single shot algorithm and the sequential active learning on MNIST with LeNet5. The acquisitions batch size of the sequential active learning were set to 100. Our algorithm ran with $\lambda = t = 0$. For the network training we used SGD with batch size 128, leaning rate 0.1 and no l2 regularization. The SGD number iterations is equivalent to 20 epochs of the full train set.

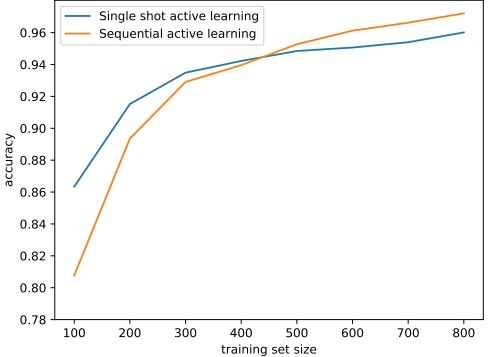

Figure 5: Single shot active learning vs sequential active learning. MNIST and (standard) LeNet5

Initially, our selection procedure shows a clear advantage. However, once the training set grows large enough, the benefit of a sequential setup starts to kick-in, the sequential algorithm starts to show superior results. This experiment motivates further development of sequential version of our algorithm.

