# OpenReview forum: "Experimental Design for Overparameterized Learning with Application to Single Shot Deep Active Learning"
_ICLR.cc/2021/Conference — Reject_

### Official Review · AnonReviewer3 · 2020-10-27
**Reasonable analysis, but technical significance is somewhat weak**

**Rating:** 5
**Confidence:** 3

**Review:**

The paper proposes an active learning (AL) strategy that is based on both of variance and bias of the linear model unlike the classical variance-based criteria. The paper claims that, for over-parameterized setting, incorporating the bias information is particularly important, and further, shows a kernelized extension and a greedy calculation strategy.

The paper focuses on an important problem of active learning, and is written clearly. The approach is reasonable, and the method would be easy to implement. However, technical significance would be somewhat weak. The derived bound of the expected error is quite simple, though it is ok but an important unknown constant is kept and no detailed discussion is provided for that constant. Further, the experimental result is not convincing to show the superiority of the proposed criterion compared with other existing approaches. Detailed comments are as follows.

In the first section, the authors mentioned about the recent 'double decent' phenomenon in over-parameterized model, and related the proposed approach with it. However, I couldn't find any particular technical procedures or discussions specific to the double decent phenomenon in the main text. Although the authors claimed that the proposed method can mitigate the double descent phenomena, this claim is not so particularly attractive currently because the classical OED also mitigates it in the illustration of Figure 1. If the proposed method can mitigate double decent that classical OED cannot, it would be attractive.

The derivation of the proposed AL criterion is performed through the classical expected risk. The paper repeatedly claims importance of incorporating bias-dominated or mixed nature, unlike classical variance based approach. However, this idea (considering the bias effect in AL) itself is not novel though it is not clearly described. For example, 'Francis R. Bach, Active learning for misspecified generalized linear models, NeurIPS2007'.

A similar bound analysis to the paper is also shown in 'Gu, et al., Selective Labeling via Error Bound Minimization, NeurIPS2012'. Although this paper is for a manifold based semi-supervised learning, it includes the usual linear model estimation as a special case. They derived a bound for the estimation error of the parameter that reflects both of the bias and variance terms (from which expected error can be derived). Their approach is similar to (Yu et al 2006) that the authors cited, but an important point is that they showed the criterion of (Yu et al 2006) can be seen as a bound of 'both of' bias and variance terms though it is originally shown as a variance-based. This would be important past analysis closely related to the proposed method, and should have been mentioned.

Obviously, hyper-parameter t would be a key factor in the proposed criterion. However, no practical discussion on how to set t is provided. In my opinion, as far as this constant remains unknown, the proposed bound is not so particularly innovative. Further the regularization parameter lambda can be problematic because unlike the usual supervised learning scenario, it sometimes should be determined before observing y.

The experiment in Section 7 is not convincing because the authors only compared with random design, and it only shows results of the bias-dominated setting of the proposed method, though the mixed nature of the expected error was repeatedly emphasized in the paper. Comparison with other state-of-the-art AL or classical OED would be indispensable.

---

> ### Author Response · Authors · 2020-11-23
> **Response to the issues raised by the reviewer**
>
> We thank the reviewer for his constructive comments,
>
>
> (1) “I couldn't find any particular technical procedures or discussions specific to the double decent phenomenon in the main text. Although the authors claimed that the proposed method can mitigate the double descent phenomena, this claim is not so particularly attractive currently because the classical OED also mitigates it in the illustration of Figure 1”
>
> The technical procedure appears after Prop 3.  We slightly revised the next in order the clarify the connection. What we try to express in Figure 1 is that while classical methods mitigate the double decent issue, they do it on the expense of not dealing well with the bias, as can be seen in Figure 1: the bias oriented regime is not handled well by classical methods. Yes, they flatten the variance dominated area, but do not gain anything in the bias dominated area.
>
> (2) It’s true that 'Francis R. Bach, Active learning for misspecified generalized linear models, NeurIPS2007' also relates to the bias term. However, the bias term has a different source in their work (misspecified model) and the solution (re-waiting the loss) is fundamentally different than ours. We were aware of this work, but failed to cite it do to oversight. We now added a citation.
>
> (3) Regarding the work of 'Gu, et al., Selective Labeling via Error Bound Minimization, NeurIPS2012': we were actually not aware of it and we thank the reviewer for pointing us to it. Indeed, Gu, et al. showed that the criterion of Yu et al 2006 is a bound on both the bias and variance, however they did not show what we show in Prop 4., that  the criterion of Yu et al 2006 is actually equal to ours when lambda=t. We think that this fact is crucial and has practical meanings. We were already related to this practical meaning in Appendix D.  However, in the revised version we clarified it further, with direct reference to the work of  'Gu, et al., Selective Labeling via Error Bound Minimization, NeurIPS2012' and a new experiment that show why the  'Gu, et al./Yu et al.  it is important to understand that the bound implies t=lambda and how to use it.
>
> (4) Obviously a good setting of the parameter t requires some good model of the noise (that what we see in figure 1. actually). In sequential experimental design there is plenty of room to set both t and lambda. However, we stress that even in single shot experimental design there is setting then t=lambda as Yu et al/Gu, et al., suggest may. We made it more clear now with a new experiment on UCI data that shows that many times setting t=0 is better then setting t=lambda, and anyway is not particularly worse.
>
>  (5)
> The experiment section is now improved. There is a direct comparison with what we see as our direct competitors Yu et al. and Core Set (Sener & Savarese, 2018), We show new state of the art results on UCI classification tasks and add the comparison to Core Set to Fig 2 comparing to the single shot deep active learning.

---

### Official Review · AnonReviewer1 · 2020-10-28
**limited technical novelity, weak experiments**

**Rating:** 3
**Confidence:** 4

**Review:**

This paper studies a straightforward generalization of v-optimality from linear regression to (kernel) ridge regression. A standard greedy algorithm is used to optimize the v-optimality criterion.  A simple experiment is conducted comparing the proposed method with random sampling on MNIST.

I vote for rejection. This is a simple derivation of the v-optimality of ridge regression (or Bayesian linear regression). The novelty is somewhat limited. For a paper with such limited technical novelty, the empirical studies are too thin; only one experiment is presented, comparing with only a naive random baseline.

The title “single shot active learning” seems a little inappropriate. As far as I understand, the word “active” in the context of active learning means the model “actively” query labels from some oracle in an __iterative__ fashion, so it already means “sequential” I think. Also, it's easy for people to confuse this with "one-shot learning".

typo: K=VV^t \in R^{n,n} should be {m,m}?

---

> ### Author Response · Authors · 2020-11-23
> **Response to the issues raised by the reviewer**
>
> We thank the reviewer for his constructive comments,
>
> As for the technical novelty - Our work reveals an important connection between our criterion and the two of what we see as the main methods in this field: Transductive Experimental Design (Yu et al. 2006) and Core Set  (Sener & Savarese, 2018) (See Prop 3,4, Appendix B).
>
>
> As for the experiments -  We compared ourselves to Core Set and to Transductive Experimental Design. This comparison is now made clearer with a new experiment on UCI classification datasets in which we show state of the art results. Furthermore, the results of the comparison to coresets was moved from the appendix to  Figure 2.
>
>
> As for the title “single shot active learning” - In many articles, Optimal Experimental Design and Active Learning are synonyms terms, where the first one is used by the ML community and second by the statistical community. Both of them do not have to be sequential, per our understanding, and classical experimental design is definitely “single shot”. We use the term “single shot active learning” and not “single shot experimental design” because it is more suitable for label prediction tasks (and not for parameter estimation for example).
>
> We fixed the typo, and thank the reviewer for notifying us about it.

---

### Official Review · AnonReviewer4 · 2020-10-29
**Interesting but incomplete work**

**Rating:** 4
**Confidence:** 3

**Review:**

This authors analyzed the V-optimality criterion of the experimental design for the ridge regression setting, proposed a greedy algorithm to optimize the criterion, and connected with the infinite neural tangent kernel.

The paper contains a number of interesting results, particularly in connecting the V-optimality criterion with the bias-variance trade-off in predictive modeling. However, the problem setting has some overly strong assumptions, such as prefixed design and iid errors, which can make the solutions trivial and not very useful in practice.

I don't recommend the paper to be accepted in its current form for the following reasons:

1. I think the main results of the paper in Section 3-5 concerns a conventional problem: Experimental design for the ridge regression. They do not provide a solution aligned with the main motivation, bridging experimental design and deep learning.
2. The theoretical analysis did not provide sufficient new insights. It actually confuses me that it introduces an additional parameter t on page 5, without exploring its relationship with the original regularization parameter lambda.
3. The connection with deep active learning is not sufficiently explained. Also the evaluation task looks overly simple and uses a trivial baseline; I would expect at least a comparison with another active learning algorithm, for example, using the coreset approach.

Other specific concerns:

-  I find the expression "single shot active learning" confusing. Does it mean active learning with a prefixed (hence unsupervised) design?
- The last sentence of Sec 2, para 1 is incorrect. Actually, it contradicts Eq 1.


In general, I think that the paper can be improved with a more focused contribution statement, and better connecting the results with the claimed contributions. The paper's relevance of deep active learning is unconvincing to me; probably the results are more suitable for a different conference.

---

> ### Author Response · Authors · 2020-11-23
> **Response to the concerns raised by the reviewer**
>
> We thank the reviewer for his constructive comments. We focus on the reasons he gave for rejecting the paper.
>
> 1) We first try to examine active learning in an overparameterized linear model and then see the relation of it with deep active learning. As two first results in this direction we have:
> (a) A theoretical analysis which shows that the popular core set approach for deep active learning approach might be suboptimal (see Prop 3 and Appendix B);
> (b) A novel successful single shot active learning /experimental design for deep learning (see Fig 2.)
> We also experimentally support our conjecture that this approach can further lead to a novel sequential algorithm but we leave it to further research.
> 2) We actually explored the relation of t with lambda (see Prop 4.)
> 3) We have experimental results with core set. Originally they were  reported in the appendix, but we now moved them to Fig. 2.
>
> Regarding the term "single shot active learning":  the meaning is that there is no opportunity to alternate between improving the model and acquiring new labels, they all have to be acquired in one batch before starting to train. We will try to make it clearer in the paper.
>
> We thank the reviewer for putting our attention to the typo in sentence (1).  It is fixed now.

---

### Official Review · AnonReviewer2 · 2020-10-30
**Good development. Unconvincing results. Limited applications.**

**Rating:** 4
**Confidence:** 4

**Review:**

In this paper, the authors develop a data selection scheme aimed to minimize a notion of Bayes excess risk for overparametrized linear models. The excess Bayes risk is the expected squared error between the prediction and the target. The authors note that solutions such as V-optimality exist for the underparametrized cases (linear regression), and offer extensions to ridge regression. After the development of a greedy schemes and a tentative extension to deep learning models, the authors show that their selection scheme can outperform random selection on MNIST with a specific model.

Active learning, both online and batch, is a well-studied problem with real-world applications. The paper does a fair job of developing the new technique but the addition seems incremental, and the algebra even though involved, is largely straightforward. The results are also unconvincing, and the one figure in the main body (Fig 2) only shows comparison with a weak (random selection) baseline. Furthermore, in the deep learning regime, 500-800 data points are somewhat unrealistic and do not cement the value of the method in real-world scenarios. Some other concerns include:

1. Unsubstantiated claims like “As such, the theory reported in the literature is often not applicable in the interpolative regime”.

2. Unconvincing contributions description, “sometimes able to mitigate the double descent phenomena”, “sometimes able to find better design than random selection”.

3. The computational complexity of the method is prohibitive — O(m^2 n^2), where m is the dataset size.

4. Minor formatting issues: “revealing a three possible regimes”, “an interesting connections”, “approach that suggest differs”, “larger then”, “expected expected risk”, etc.

Overall, even though active learning is a highly relevant problem domain, in my opinion, the paper falls short in establishing itself as a strong competitor in the domain for reasons described previously. I would encourage the authors to resubmit with stronger justifications of the contribution and more convincing results.

---

> ### Author Response · Authors · 2020-11-23
> **Response to the concerns raised by the reviewer**
>
> We thank the reviewer for  his constructive comments.
>
> 1) “As such, the theory reported in the literature is often not applicable in the interpolative regime”. The fact that classical experimental design works in the under-parameterized regime can be verified by reading classical work which we cite in the related work section. The fact that the most of the classical work is not applicable for the overparameterized regime is explained in section 2.
>
> 2) “sometimes able to find better design than random selection” - We changed it to “our algorithm is able to find designs that are better than state of the art”. That statement is based on comparison with our two direct competitors - Transductive Experimental Design (Yu et al.)  and Core Set (Sener & Savarese, 2018). The comparison with Core Set was originally reported in the appendix, and we did not put it in Fig 2. However, we now added it to the main text. We also added a new important experiment on UCI classification  datasets in which we compare our method with t=0 vs Yu et al. 2006 , which is a special case of our criterion (t=lambda) as we show in Prop 4.
>
> 3) We stress that reducing the number of samples from ten of thousands to hundreds is relevant for real life deep learning tasks.  For example consider sequential batch active learning. Our work is the first work to suggest solution for how to choose the first batch.

---

### Author Response · Authors · 2020-11-23
**Main revisions in the final version**

We thank the reviewers for their very enlightening comments.

Our main reversions for the final version of the paper are as follows:

1) We added a new experiment that shows state-of-the-art results on UCI classification datasets*.

State-of-the-art in the sense that  we show better result than what we see with our direct competitor in this field: transductive experimental design (Yu et al. 2016).

2) We added a comparison to core set in Figure 2.
3) We added an important explanation for the relation of our work to Gu, et al. 2012 (end of Sec 3)
4) We revised a little bit the contribution section, clarifying things and highlighting our contributions in light of recent experiments.

---

### Decision · Program_Chairs · 2021-01-07
**Final Decision**

**Decision:**

Reject

**Comment:**

This paper gives a method of performing experimental design (one-round active learning) in overparameterized regression. Although the comparison with the coreset method baseline is a nice addition, the reviewers still have concerns in the following aspects:
- The novelty compared to the classical v-optimality design is limited
- The hyperparameter t is hard to choose in practice. This is important because each different hyperparameter setting would induce a different set of examples for label queries.
- Computational complexity
- It is unclear exactly how the proposed method (or other experimental design method) can mitigate double descent

We encourage the authors to take these into account in the revision.